# Efficient hydrogen production on MoNi$_4$ electrocatalysts with fast water dissociation kinetics

Jian Zhang[1], Tao Wang[2], Pan Liu[3,4], Zhongquan Liao[5], Shaohua Liu[1], Xiaodong Zhuang[1], Mingwei Chen[3,4], Ehrenfried Zschech[5] & Xinliang Feng[1]

Various platinum-free electrocatalysts have been explored for hydrogen evolution reaction in acidic solutions. However, in economical water-alkali electrolysers, sluggish water dissociation kinetics (Volmer step) on platinum-free electrocatalysts results in poor hydrogen-production activities. Here we report a MoNi$_4$ electrocatalyst supported by MoO$_2$ cuboids on nickel foam (MoNi$_4$/MoO$_2$@Ni), which is constructed by controlling the outward diffusion of nickel atoms on annealing precursor NiMoO$_4$ cuboids on nickel foam. Experimental and theoretical results confirm that a rapid Tafel-step-decided hydrogen evolution proceeds on MoNi$_4$ electrocatalyst. As a result, the MoNi$_4$ electrocatalyst exhibits zero onset overpotential, an overpotential of 15 mV at 10 mA cm$^{-2}$ and a low Tafel slope of 30 mV per decade in 1 M potassium hydroxide electrolyte, which are comparable to the results for platinum and superior to those for state-of-the-art platinum-free electrocatalysts. Benefiting from its scalable preparation and stability, the MoNi$_4$ electrocatalyst is promising for practical water-alkali electrolysers.

[1] Center for Advancing Electronics Dresden (Cfaed) and Department of Chemistry and Food Chemistry, Technische Universitaet Dresden, 01062 Dresden, Germany. [2] Univ Lyon, Ens de Lyon, CNRS, Université Lyon 1, Laboratoire de Chimie, UMR 5182, F-69342 Lyon, France. [3] WPI Advanced Institute for Materials Research, Tohoku University, Sendai 980-8577, Japan. [4] CREST, JST, 4-1-8 Honcho Kawaguchi, Saitama 332-0012, Japan. [5] Fraunhofer Institute for Ceramic Technologies and Systems (IKTS), 01109 Dresden, Germany. Correspondence and requests for materials should be addressed to X.F. (email: xinliang.feng@tu-dresden.de).

Growing concern about the energy crisis and the seriousness of environmental contamination urgently demand the development of renewable energy sources as feasible alternatives to diminishing fossil fuels. Owing to its high energy density and environmentally friendly characteristics, molecular hydrogen is an attractive and promising energy carrier to meet future global energy demands[1,2]. In many of the approaches to hydrogen production, the electrocatalytic hydrogen evolution reaction (HER) from water splitting is the most economical and effective route for the future hydrogen economy[3–6]. To accelerate the sluggish HER kinetics, particularly in alkaline electrolytes, highly active and durable electrocatalysts are essential to lower the kinetic HER overpotential[7,8]. As a benchmark HER electrocatalyst with a zero HER overpotential, the precious metal platinum (Pt) plays a dominant role in present $H_2$-production technologies, such as water-alkali electrolysers[9–11]. Unfortunately, the scarcity and high cost of Pt seriously impede its large-scale applications in electrocatalytic HERs.

To develop efficient and earth-abundant alternatives to Pt as HER electrocatalysts, great efforts have been made to understand the fundamental HER mechanisms on the surfaces of electrocatalysts in alkaline environments[12,13]. The HER kinetics in alkaline solutions involves two steps: electron-coupled water dissociation (the Volmer step for the formation of adsorbed hydrogen); and the concomitant combination of adsorbed hydrogen into molecular hydrogen (the Heyrovsky or Tafel step; Supplementary Note 1)[12,14]. Accordingly, the HER activity of an electrocatalyst in alkaline electrolytes is synergistically dominated by the prior Volmer step and subsequent Tafel step[13]. The low energy barrier ($\Delta G(H_2O) = 0.44$ eV) of the Volmer step provides the Pt catalyst with a fast Tafel step-determined HER process (Tafel slop = 30 mV per decade) in alkaline electrolytes, which is responsible for its excellent HER activity[12,15]. Inspired by the fundamental HER mechanism that occurs on Pt, the development of novel Pt-free electrocatalysts with a significantly accelerated Volmer step is an appealing approach. Recently, several electrocatalysts with a decreased HER overpotential, such as CoP/S (with an overpotential at 10 mA cm$^{-2}$: ~48 mV) and $Mo_2C$/graphene (with an overpotential at 10 mA cm$^{-2}$: ~34 mV) have been reported in acidic solutions[16,17]. Nevertheless, under alkaline conditions, the sluggish Volmer step on these Pt-free electrocatalysts results in far lower HER activity than the Pt catalyst[18–21].

In past decades, various Ni- or Mo-based oxides, hydroxides, layered double hydroxides, phosphides and sulfides have been reported as electrocatalysts for water splitting. Ni atoms are broadly recognized as excellent water dissociation centres, while Mo atoms have superior adsorption properties towards hydrogen[13,22–24]. Therefore, Mo–Ni-based alloy electrocatalysts ($Mo_xNi_y$) can be promising candidates to effectively reduce the energy barrier of the Volmer step and speed up the sluggish HER kinetics under alkaline conditions. In this study, we demonstrate

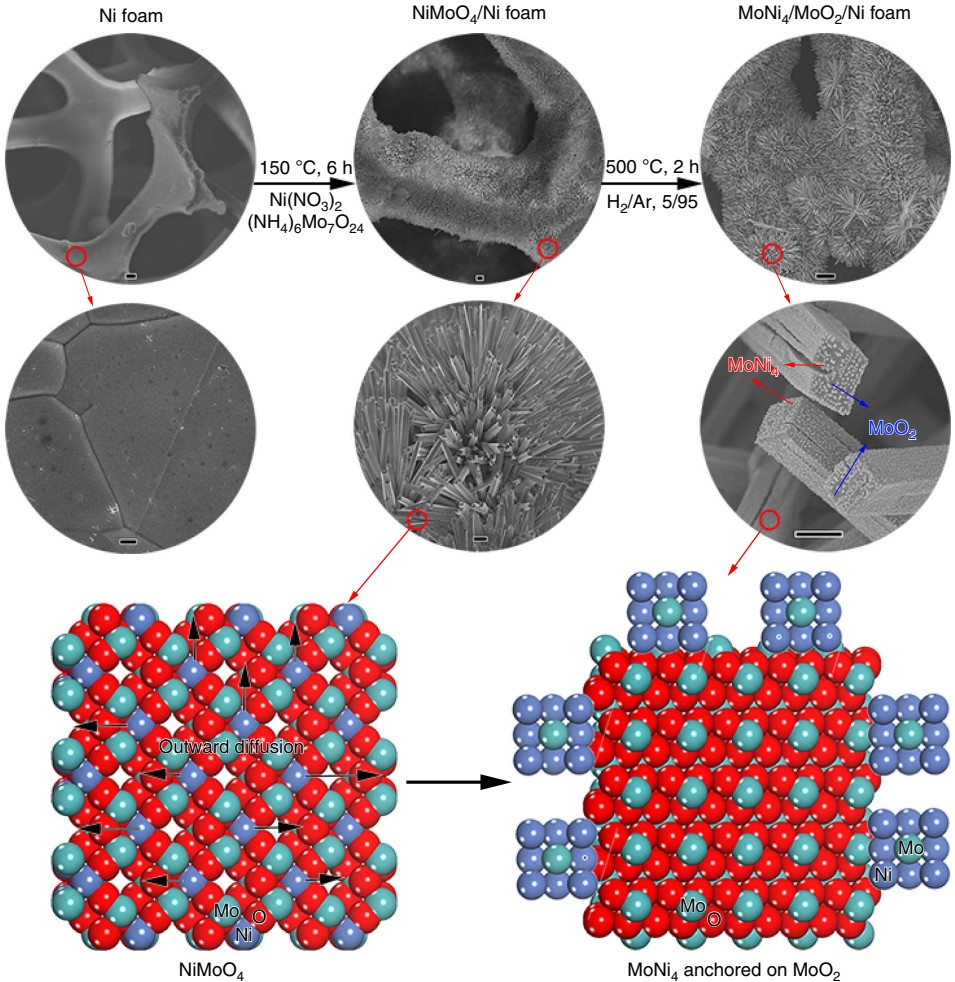

**Figure 1 | Synthetic scheme of MoNi4 electrocatalyst supported by the MoO2 cuboids on nickel foam.** Synthetic scheme of $MoNi_4$ electrocatalyst supported by the $MoO_2$ cuboids on nickel foam. Scale bars, Ni foam, 20 μm (top) and 1 μm (bottom); $NiMoO_4$/Ni foam, 10 μm (top) and 2 μm (bottom); $MoNi_4$/$MoO_2$/Ni foam, 20 μm (top) and 1 μm (bottom).

a MoNi$_4$ electrocatalyst anchored on MoO$_2$ cuboids, which are vertically aligned on nickel foam (MoNi$_4$/MoO$_2$@Ni). MoNi$_4$ nanoparticles with a size of 20–100 nm are constructed *in situ* on the MoO$_2$ cuboids by controlling the outward diffusion of Ni atoms when previously synthesized NiMoO$_4$ cuboids are heated in a H$_2$/Ar (v/v, 5/95) atmosphere at 500 °C. The resultant MoNi$_4$/MoO$_2$@Ni exhibits a high HER activity with a zero onset overpotential and a low Tafel slope of ~30 mV per decade in a 1 M KOH aqueous solution, which are highly comparable to those for the Pt catalyst (onset overpotential: 0 mV; Tafel slope: 32 mV per decade). In addition, the achieved MoNi$_4$ electrocatalyst requires low overpotentials of only ~15 and ~44 mV to stably deliver cathodic current densities of 10 and 200 mA cm$^{-2}$, respectively, presenting state-of-the-art HER activity amongst all reported Pt-free electrocatalysts[7,10,18]. Experimental investigations reveal that the MoNi$_4$ electrocatalyst behaves as the highly active centre and manifests fast Tafel step-determined HER kinetics. Furthermore, density functional theory (DFT) calculations determine that the kinetic energy barrier of the Volmer step for the MoNi$_4$ electrocatalyst is as low as 0.39 eV. These results confirm that the sluggish Volmer step is drastically accelerated for the MoNi$_4$ electrocatalyst.

## Results

**Synthesis of the MoNi$_4$ electrocatalyst**. The synthesis of the MoNi$_4$ electrocatalyst involves two steps, as illustrated in Fig. 1. First, the NiMoO$_4$ cuboids were grown beforehand on a piece of nickel foam ($1 \times 3$ cm$^2$) via a hydrothermal reaction at 150 °C for 6 h in 15 ml of deionized water containing Ni(NO$_3$)$_2$•6H$_2$O (0.04 M) and (NH$_4$)$_6$Mo$_7$O$_{24}$•4H$_2$O (0.01 M). Second, when the as-synthesized NiMoO$_4$ cuboids were calcined in a H$_2$/Ar (v/v, 5/95) atmosphere at 500 °C for 2 h, the inner Ni atoms diffused outward due to the formation of MoO$_2$. As a result,

MoNi$_4$ nanoparticles were directly constructed on the surfaces of the MoO$_2$ cuboids. To probe the formation mechanism of the MoNi$_4$ nanoparticles, different calcination temperatures and times were investigated (Supplementary Figs 1–5). In comparison with the smooth surfaces of precursor NiMoO$_4$ at 400 °C, the appearance of numerous surface nanoparticles at 500 °C indicated the formation of MoNi$_4$ on the resulting MoO$_2$ cuboids (Supplementary Fig. 1a,b). When the calcination temperature reached 600 °C, MoNi$_3$ nanoparticles on the MoO$_2$ cuboids (MoNi$_3$/MoO$_2$@Ni) were produced due to the continuous reduction of MoO$_2$ (Supplementary Fig. 1c,d). In addition, with increased calcination time at 500 °C, the MoNi$_4$ nanoparticles gradually emerged and grew into bulk particles on the MoO$_2$ cuboids (Supplementary Figs 2–5).

**Structural characterizations of the MoNi$_4$ electrocatalyst**. X-ray diffraction characterization reveals that the crystalline structure of the as-obtained precursor on the Ni foam can be indexed to NiMoO$_4$ (Supplementary Fig. 6). The morphology of NiMoO$_4$ was scrutinized by scanning electron microscopy (SEM). As shown in Supplementary Figs 7 and 8, dense NiMoO$_4$ cuboids with sizes in the range of 0.5–1.0 μm and lengths of tens of microns are vertically aligned on the nickel foam. Elemental mapping, energy dispersive spectroscopy and X-ray photoelectron spectroscopy (XPS) confirm that the NiMoO$_4$ cuboids consist of Ni, Mo and O elements, and the molar ratio of Ni to Mo is ~1:1.01 (Supplementary Figs 9 and 10).

The product of the NiMoO$_4$ cuboids on the Ni foam calcined at 500 °C for 2 h was surveyed with X-ray diffraction using Cu-Kα radiation, SEM and high-resolution transmission electron microscopy (HRTEM). In Supplementary Fig. 11, the sharp X-ray diffraction diffraction peaks at ~44.6°, 52.0° and 76.5° originate from the Ni foam (JCPDS, No. 65–2865). The peaks located at

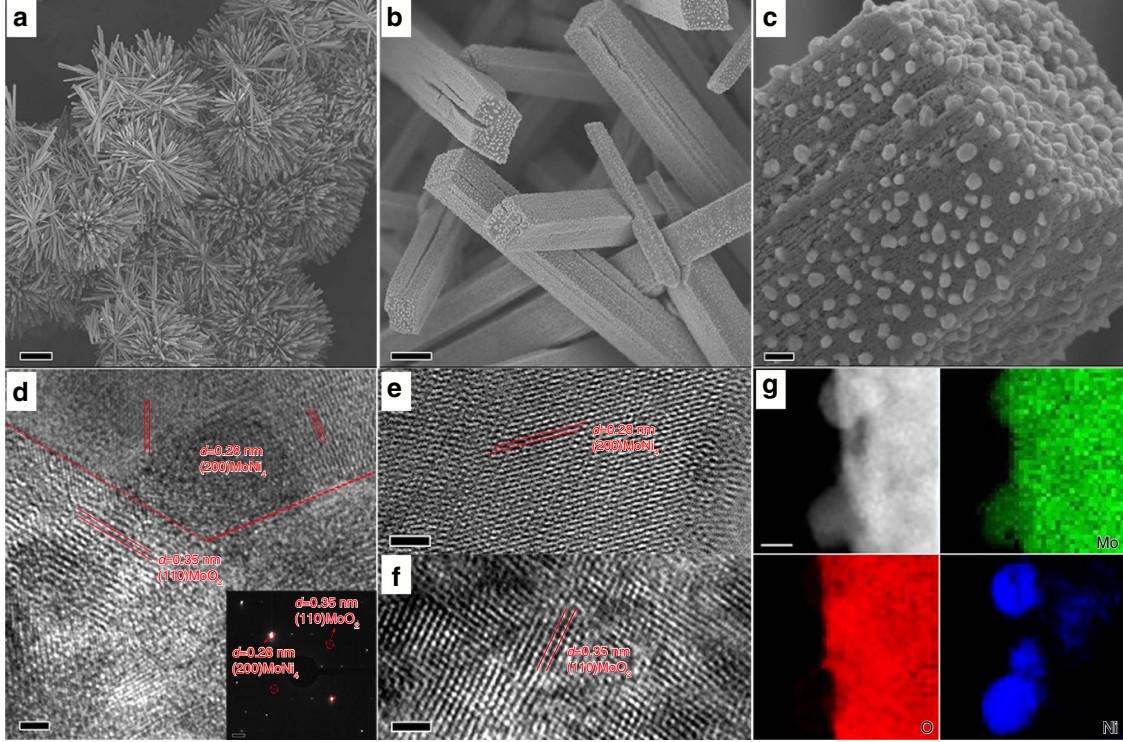

**Figure 2 | Morphology and chemical composition analyses of MoNi$_4$/MoO$_2$@Ni.** (**a–c**) Typical SEM and (**d–f**) HRTEM images of MoNi$_4$/MoO$_2$@Ni; (**g**) corresponding elemental mapping images of the MoNi$_4$ electrocatalyst and the MoO$_2$ cuboids. The inset image in **d** is the related selected-area electron diffraction pattern of the MoNi$_4$ electrocatalyst and the MoO$_2$ cuboids. Scale bars, (**a**) 20 μm; (**b**) 1 μm; (**c**) 100 nm; (**d–f**) 2 nm; inset in **d**, 1 1/nm; (**g**) 20 nm.

26.3°, 37.0°, 41.5°, 49.5°, 53.7°, 60.5° and 66.9° are indexed to metallic $MoO_2$ (JCPDS, No. 32-0671). The appearance of peaks at 31.0° and 43.5° are assigned to the (200) and (121) facets of $MoNi_4$ (JCPDS, No. 65–5480), respectively. Thus, these result suggest that the obtained product on the nickel foam consists of $MoNi_4$ and $MoO_2$. As shown in Fig. 2a–c, numerous nanoparticles with sizes in the range of 20–100 nm are uniformly anchored on the cuboids, which are vertically aligned on the nickel foam. The corresponding energy-dispersive X-ray spectroscopy (EDX) analysis further confirms that the products are composed of Mo, Ni and O, and the molar ratio of Mo to Ni is ∼1:1.3 (Supplementary Fig. 12). Clearly, the HRTEM images of the samples show lattice fringes with lattice distances of 0.35 and 0.28 nm, which correspond to the (110) facet of $MoO_2$ and the (200) facet of $MoNi_4$, respectively (Fig. 2d–f). The selected-area electron diffraction pattern shows diffraction patterns of the (200) facet of $MoNi_4$ and the (110) facet of $MoO_2$ (the inset in Fig. 2d). Noticeably, the scanning TEM–EDX characterizations indicate that the surface nanoparticles are constituted by only Mo and Ni with an atomic ratio of 1:3.84, which well approaches to 1:4 (Fig. 2g and Supplementary Fig. 13). The XPS analysis was carried out to probe the chemical compositions and surface valence states of the $MoNi_4$ nanoparticles and the supporting $MoO_2$ cuboids. As illustrated in Supplementary Fig. 14, the XPS spectrum confirms the presence of Mo, Ni and O, and the molar ratio of Mo to Ni is ∼1:1.1. As shown in Supplementary Figs 15–17, XPS peaks of metallic $Mo^0$ and $Ni^0$ are observed at 229.3 and 852.5 eV, respectively, further confirming the existence of $Mo^0$ and $Ni^0$ in the surfaces of $MoNi_4/MoO_2@Ni$.

**Electrocatalytic HER performance.** To evaluate the electrocatalytic HER activities of the electrocatalysts, a three-electrode system in an Ar-saturated 1 M KOH aqueous solution was used using a Hg/HgO electrode and a graphite rod as the reference and counter electrodes, respectively (Supplementary Fig. 18). All potentials are referenced to the reversible hydrogen electrode (RHE), and the ohmic potential drop loss from the electrolyte resistance has been subtracted (Supplementary Figs 19 and 20). For comparison, pure Ni nanosheets and $MoO_2$ cuboids were also prepared on the nickel foam using the hydrothermal reactions (Supplementary Figs 21–25). As displayed in Fig. 3a and Supplementary Fig. 26, a commercial Pt/C electrocatalyst deposited on the nickel foam (weight density: $1\ mg\ cm^{-2}$) using Nafion as a binder exhibited a zero HER onset overpotential and delivered a current density of $10\ mV\ cm^{-2}$ at an overpotential of ∼10 mV. However, the maximum current density only reached $80\ mA\ cm^{-2}$ due to the Pt catalyst significantly peeling off from the support, caused by the generated $H_2$ bubbles. Although the Ni nanosheets on the nickel foam could act as an HER electrocatalyst, the HER occurred at a very high overpotential of ∼253 mV. For the $MoO_2$ cuboids on the nickel foam, the cathodic current density of $10\ mA\ cm^{-2}$ was delivered at an overpotential as large as ∼48 mV. In comparison to the Ni nanosheets and the $MoO_2$ cuboids, the $NiMoO_4$ cuboids and $MoNi_3/MoO_2$ cuboids on the nickel foam exhibited a similar onset overpotential of ∼10 mV and an overpotential of ∼30 and 37 mV at $10\ mA\ cm^{-2}$, respectively (Supplementary Figs 27–29). Remarkably, $MoNi_4/MoO_2@Ni$ exhibited an onset overpotential of 0 mV, which was highly comparable to that of the Pt catalyst. In addition, for the supported $MoNi_4$ electrocatalyst, the overpotential at current densities of 10 and $200\ mA\ cm^{-2}$ was as low as ∼15 and 44 mV, respectively, which were significantly lower than the values for the Ni nanosheets, $MoO_2$ cuboids, $NiMoO_4$ cuboids, $MoNi_3/MoO_2$ cuboids and state-of-the-art Pt-free HER electrocatalysts such as NiO/Ni heterostructures (∼85 mV

at $10\ mA\ cm^{-2}$)[25], pyrite-type CoPS nanowires (∼48 mV at $10\ mA\ cm^{-2}$)[16], nickel doped carbon (∼34 mV at $10\ mA\ cm^{-2}$)[26], a $Mo_2C$/carbon/graphene hybrid (∼34 mV at $10\ mA\ cm^{-2}$)[17], $MoSSe/NiSe_2$ foam (∼69 mV at $10\ mA\ cm^{-2}$)[27], $Fe_{0.9}Co_{0.1}S_2$/carbon nanotubes (∼100 mV at $10\ mA\ cm^{-2}$)[28], $Ni_2P$ nanoparticles (∼120 mV at $10\ mA\ cm^{-2}$)[29] and strained $MoS_2$ nanosheets (∼170 mV at $10\ mA\ cm^{-2}$)[30] (Supplementary Table 1)[31–39].

Figure 3b displays the Tafel plots of the corresponding polarization curves, which provide profound insights into the fundamental HER kinetic mechanism occurring on the surfaces of the electrocatalysts. As a result of the low energy barrier (0.44 eV on Pt) of the Volmer step, the kinetic rate-limiting step for the Pt catalyst is the Tafel process, and the theoretical Tafel slope is 30 mV per decade (here the Tafel slope of the commercial Pt catalyst was measured to be 32 mV per decade)[12]. Remarkably, the Tafel slope of the $MoNi_4$ electrocatalyst was as low as 30 mV per decade, which is far lower than the values of 129 mV per decade for the Ni nanosheets and 75 mV per decade for the $MoO_2$ cuboids and highly comparable to that of the Pt-based catalyst (Fig. 3c and Supplementary Table 1). This result indicated that the electrocatalytic HER kinetics on the $MoNi_4$ electrocatalyst were determined by the Tafel step rather than a coupled Volmer–Tafel or Volmer–Heyrovsky process. In other words, the prior Volmer step has been significantly accelerated. The exchange current density of the $MoNi_4$ electrocatalyst was estimated to be ∼$1.24\ mA\ cm^{-2}$ (Supplementary Fig. 30). To clarify the influence of the active surface area on the electrocatalytic HER activity, the corresponding electrochemical double-layer capacitances (Cps) of the electrocatalysts were analysed by applying cyclic voltammetry cycles at different scan rates[40]. The Cps of the Ni nanosheets and $MoO_2$ cuboids were ∼0.001 and 0.640 F, respectively, while the $MoNi_4$ electrocatalyst had a high Cp of 2.220 F (Supplementary Fig. 31). On the basis of its Cp, the $MoNi_4$ electrocatalyst was calculated to have a turnover frequency of $0.4\ s^{-1}$ at a low overpotential of 50 mV, which was higher than the turnover frequency values of the previously reported Pt-free electrocatalysts (Supplementary Fig. 32 and Supplementary Table 1)[41–44].

Long-term electrocatalytic stability is another important criterion for HER electrocatalysts. To investigate the durability of the $MoNi_4$ electrocatalyst, continuous cyclic voltammetry scans were performed between 0.2 and −0.2 V at a scan rate of $50\ mV\ s^{-1}$ in a 1 M KOH solution. As depicted in Fig. 3d, the HER overpotential of the $MoNi_4$ electrocatalyst at $200\ mA\ cm^{-2}$ increased by only 6 mV after 2,000 cyclic voltammetry cycles. In addition, a long-term electrocatalytic HER process was successively carried out at current densities of 10, 100 and $200\ mA\ cm^{-2}$ (Supplementary Movie 1). The inset in Fig. 3d demonstrates that the $MoNi_4$ electrocatalyst retained a steady HER activity, and only an increase of ∼3 mV in potential was observed at a current density of $10\ mA\ cm^{-2}$ after a period of 10 h of hydrogen production. The overpotential required for large current densities of 100 and $200\ mA\ cm^{-2}$ was augmented by only 2 and 5 mV, respectively. After a series of HER durability assessments, the structure of the $MoNi_4$ electrocatalyst was examined using SEM and HRTEM. $MoNi_4/MoO_2@Ni$ showed no structural variations, highlighting the superior structural robustness of the $MoNi_4$ electrocatalyst during the electrocatalytic HER process (Supplementary Figs 33–36).

The approach to the synthesis of $MoNi_4/MoO_2@Ni$ is scalable on the nickel foam. The $MoNi_4$ electrocatalyst was thus prepared on commercially available nickel foam with dimensions of $6 \times 20\ cm^2$. As shown in Supplementary Fig. 37, the $MoNi_4$ electrocatalyst supported by the $MoO_2$ cuboids on the nickel foam was free-standing and highly flexible. It is notable that the

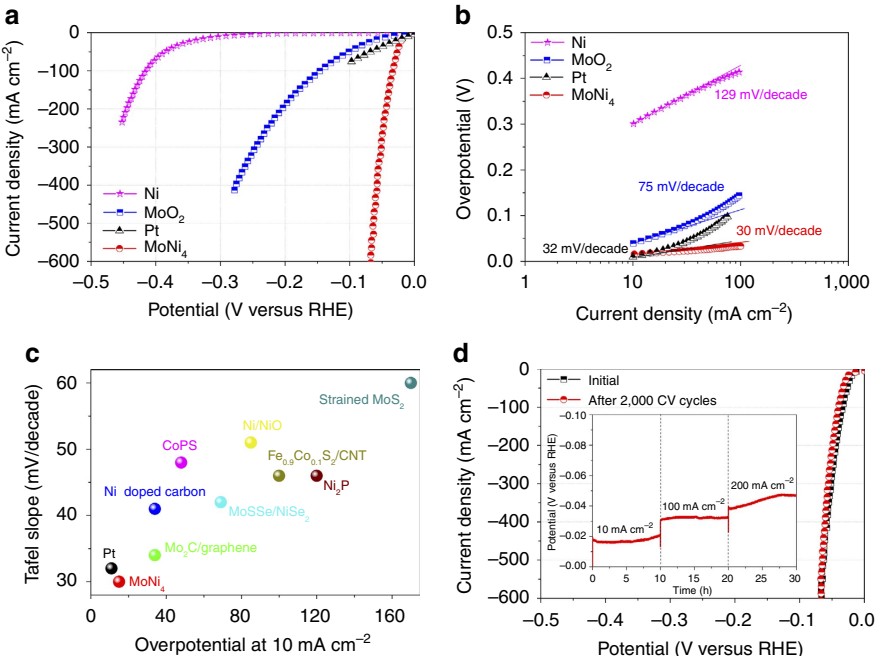

**Figure 3 | Electrocatalytic activities of different catalysts. (a)** Polarization curves and **(b)** Tafel plots of the MoNi$_4$ electrocatalyst supported by the MoO$_2$ cuboids, pure Ni nanosheets and MoO$_2$ cuboids on the nickel foam. **(c)** Comparison with selected state-of-the-art HER electrocatalysts. **(d)** Polarization curves of the MoNi$_4$ electrocatalyst before and after 2,000 cyclic voltammetry cycles; inset: long-term stability tests of the MoNi$_4$ electrocatalyst at different current densities: 10; 100; and 200 mA cm$^{-2}$. Electrolyte: 1 M KOH aqueous solution; scan rate: 1 mV s$^{-1}$.

MoNi$_4$ electrocatalyst unveiled a steady HER activity even though the supporting Ni foam was deformed to various degrees (Supplementary Fig. 38). For reported Raney nickel and nickel–molybdenum alloy electrodes, concentrated alkaline solutions (30 wt%) and high electrolyte temperatures (70 °C) are generally demanded to achieve high cathodic current densities of 200–500 mA cm$^{-2}$ (ref. 45). Here high cathodic current densities of up to 200 and 500 mA cm$^{-2}$ were delivered by the MoNi$_4$ electrocatalyst at extremely low overpotentials of ~44 and ~65 mV in a 5.3 wt% KOH solution at room temperature.

Afterward, a water-alkali electrolyser was built up in a 1 M KOH solution using MoNi$_4$/MoO$_2$@Ni as the cathode and a previously reported MoS$_2$/Ni$_3$S$_2$ hybrid as the anode (Supplementary Fig. 39)[38]. As exhibited in Supplementary Fig. 40a, for a noble metal-based Pt–Ir/C couple, a cell voltage of ~1.7 V was applied for a current density of 10 mA cm$^{-2}$. In contrast, the MoNi$_4$–MoS$_2$/Ni$_3$S$_2$ couple required a low cell voltage of only ~1.47 V to deliver a current density of 10 mA cm$^{-2}$, which is much lower than that for the noble metal-based Pt–Ir/C couple. Over 10 h of galvanostatic electrolysis at 10 mA cm$^{-2}$, the applied voltage of the MoNi$_4$–MoS$_2$/Ni$_3$S$_2$ couple had an augmentation of ~0.02 V, which is much lower than the value of 0.07 V for the Pt–Ir/C couple (Supplementary Fig. 40b). Moreover, the electrolyser with a high current density of 200 mA cm$^{-2}$ was durably driven by the MoNi$_4$–MoS$_2$/Ni$_3$S$_2$ couple at a low voltage of ~1.70 V (Supplementary Movie 2).

**HER active centres.** To understand the intrinsic contributions of the surface MoNi$_4$ nanoparticles and the underlying MoO$_2$ cuboids to the HER activity, pure MoO$_2$ nanosheets and MoNi$_4$ nanoparticles supported by MoO$_2$ cuboids were also synthesized on carbon cloth. Thus, the contribution of the underlying Ni foam could be excluded (Supplementary Fig. 41). Clearly, the pristine MoO$_2$ nanosheets on carbon cloth showed a very high

HER onset potential of ~240 mV in 1 M KOH and ~200 mV in 0.5 M H$_2$SO$_4$, suggesting that the MoO$_2$ electrocatalyst inherently presented a very sluggish Volmer step and a poor Tafel process (Supplementary Fig. 42). In contrast, the MoNi$_4$ electrocatalyst supported by the MoO$_2$ cuboids on the carbon cloth (MoNi$_4$/MoO$_2$@C) exhibited a zero onset potential, which was similar to that for MoNi$_4$/MoO$_2$@Ni. When the surface MoNi$_4$ nanoparticles of MoNi$_4$/MoO$_2$@C were etched away using 2 M H$_2$SO$_4$ aqueous solution. Obviously, the produced MoO$_2$@C showed a largely increased onset potential of ~133 mV (Supplementary Figs 43–46). These results demonstrate that the excellent HER activity of the MoNi$_4$/MoO$_2$@Ni unambiguously originates from the surface MoNi$_4$ nanoparticles rather than from the supporting MoO$_2$ cuboids.

To gain profound insight into the electrocatalytic HER active sites, we also analysed the surface electrochemical behaviour of the MoNi$_4$ electrocatalyst on the MoO$_2$ cuboids. For a freshly prepared MoNi$_4$ electrocatalyst, an electrochemical cyclic voltammetry cycle between −0.025 and 0.275 V (versus RHE) was initially performed with a scan rate of 1 mV s$^{-1}$. Obviously, the positions of the electrochemically reversible peaks shifted from 0.175 V/0.113 V to 0.215 V/0.064 V when the KOH concentration was changed from 1 to 0.1 M (Supplementary Fig. 47a). The strong dependence on the concentration of KOH as the electrolyte revealed that the electrochemically reversible peaks originated from an ad-/desorption process of water molecules or hydrogen (between 0.05 and 0.35 V, as reported) rather than from the surface redox reactions of the MoNi$_4$ electrocatalyst and supporting MoO$_2$ cuboids[12]. In addition, in contrast to the results on pure Ni nanosheets (0.150 V) and MoO$_2$ (0.164 V) cuboids, the water or hydrogen adsorption peak of the MoNi$_4$ electrocatalyst showed an anodic shift to 0.175 V, reflecting a superior water or hydrogen adsorption property (Supplementary Fig. 47b).

To evaluate the intrinsic electrocatalytic HER activity of the MoNi$_4$ electrocatalyst, the recorded cathodic current density was

normalized versus the related Brunauer Emmett Teller specific surface area of the $MoNi_4$ electrocatalyst ($32\,m^2\,g^{-1}$) (Supplementary Fig. 48). As described in Supplementary Fig. 49, when the current density was below $0.38\,A\,m^{-2}$, the polarization curve of the $MoNi_4$ electrocatalyst nearly overlapped with that of the Pt catalyst. However, the HER overpotential of the $MoNi_4$ electrocatalyst was much lower than that of the Pt catalyst at large current densities ($>0.38\,A\,m^{-2}$). These results illustrate that the intrinsic HER activity associated with the specific surface area of the $MoNi_4$ electrocatalyst is even higher than that of the Pt catalyst under alkaline conditions.

**Theoretical calculations.** To understand the fundamental mechanism of the outstanding HER activity on $MoNi_4$/ $MoO_2$@Ni, the kinetic energy barrier of the prior Volmer step ($\Delta G(H_2O)$) and the concomitant combination of adsorbed H into molecular hydrogen ($\Delta G(H)$, Tafel step) were studied using the DFT calculations according to the as-built electrocatalyst models including the (111) facet of Ni metal, the (110) facet of Mo metal, the (110) facet of $MoO_2$ and the (200) facet of $MoNi_4$ (Supplementary Fig. 50). As shown in Fig. 4, $MoO_2$ has a large energy barrier for the Volmer step ($\Delta G(H_2O) = 1.01\,eV$) and a strong hydrogen adsorption free energy ($|\Delta G(H)| = 1.21\,eV$), indicating a very sluggish Volmer–Tafel mechanism. Thus, $MoO_2$ is not the highly active centre for the HER, which agrees well with

the experimental results. The $\Delta G(H_2O)$ values on pure Ni metal and Mo metal are as high as 0.91 and 0.65 eV, respectively (Fig. 4b and Supplementary Fig. 51). In contrast, the $\Delta G(H_2O)$ on $MoNi_4$ is significantly decreased to 0.39 eV, which is even lower than the value of 0.44 eV on Pt (ref. 15). In addition, $MoNi_4$ has a lower $|\Delta G(H)|$ of 0.74 eV than the value of 1.21 eV for $MoO_2$, which corresponds to a superior hydrogen adsorption capability (Fig. 4c). Thereby, the HER reaction on $MoNi_4$ is associated with a process defined by a fast Tafel step rather than a sluggish Volmer–Tafel step (Supplementary Fig. 52).

## Discussion

In summary, we have demonstrated a $MoNi_4$ electrocatalyst supported by $MoO_2$ cuboids on nickel foam or carbon cloth. As favoured by a largely reduced energy barrier of the Volmer step, the achieved $MoNi_4$ electrocatalyst exhibits a high HER activity under alkaline conditions, which is highly comparable to that for Pt and outperforms any reported results for Pt-free electrocatalysts, to the best of our knowledge. Moreover, the large-scale preparation and excellent catalytic stability provide $MoNi_4$/ $MoO_2$@Ni with a promising utilization in water-alkali electrolysers for hydrogen production. Therefore, the exploration and understanding of the $MoNi_4$ electrocatalyst provide a promising alternative to Pt catalysts for emerging applications in energy generation.

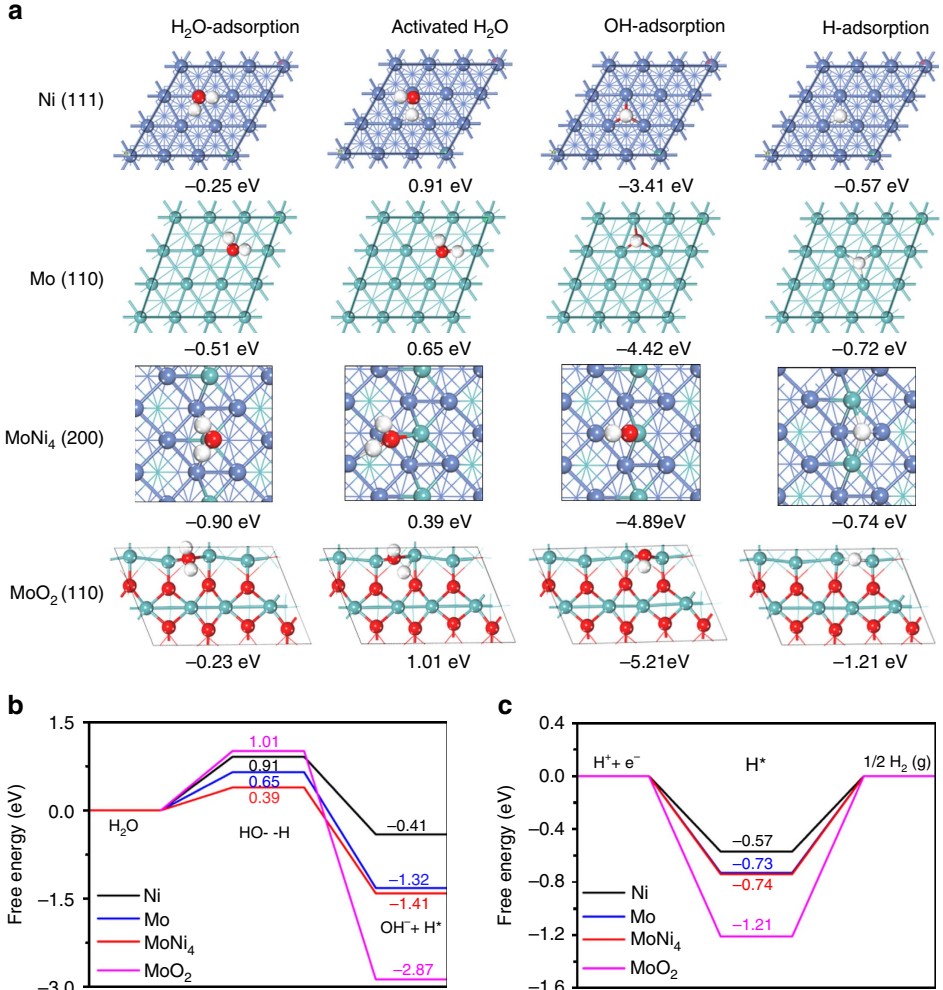

**Figure 4 | DFT calculations.** (**a**) Calculated free energies of $H_2O$ adsorption, activated $H_2O$ adsorption, OH adsorption and H adsorption. (**b**) Calculated adsorption free energy diagram for the Volmer step. (**c**) Calculated adsorption free energy diagram for the Tafel step. Blue balls: Ni; aqua balls: Mo; red balls: O.

## Methods

**Material synthesis.** To synthesize the $MoNi_4$ electrocatalyst, $NiMoO_4$ cuboids were first constructed on nickel foam through a hydrothermal reaction[46]. First, the commercial nickel foam was successively washed with ethanol, a 1 M HCl aqueous solution and deionized water. Second, one piece of nickel foam ($1 \times 3\ cm^2$) was immersed into 15 ml of $H_2O$ containing $Ni(NO_3)_2 \bullet 6H_2O$ (0.04 M) and $(NH_4)_6Mo_7O_{24} \bullet 4H_2O$ (0.01 M) in a Teflon autoclave. Third, the autoclave was heated at 150 °C for 6 h in a drying oven. After washing with deionized water, the $NiMoO_4$ cuboids were achieved on the nickel foam. Finally, the as-constructed $NiMoO_4$ cuboids were heated at 500 °C for 2 h in a $H_2/Ar$ (4:96) atmosphere, and then, the $MoNi_4$ electrocatalyst anchored on the $MoO_2$ cuboids was obtained. The loading weight of the formed $MoNi_4$ nanoparticles and $MoO_2$ cuboids on the nickel foam was $\sim 43.4\ mg\ cm^2$. The pure Ni nanosheets and $MoO_2$ cuboids on the nickel foam, as well as the pure $MoO_2$ nanosheets and $MoN_4$ nanoparticles supported by $MoO_2$ cuboids on carbon cloth, were also prepared following the same procedure for $MoNi_4$ by changing the precursors and substrates.

**Structure characterizations.** SEM, as well as corresponding elemental mapping, and EDX analysis were carried out with a Gemini 500 (Carl Zeiss) system. HRTEM was performed using a LIBRA 200 MC Cs scanning TEM (Carl Zeiss) operating at an accelerating voltage of 200 kV. XPS experiments were carried out on an AXIS Ultra DLD (Kratos) system using Al Kα radiation. XRD patterns were recorded on a PW1820 powder diffractometer (Phillips) using Cu-Kα radiation. The electrochemical tests were carried out on WaveDriver 20 (Pine Research Instrumentation) and CHI 660E Potentiostat (CH Instruments) systems.

**Electrochemical measurements.** All electrochemical tests were performed at room temperature. The electrochemical HER was carried out in a three-electrode system. A standard Hg/HgO electrode and a graphite rod were used as the reference and counter electrodes, respectively. The Hg/HgO electrode was calibrated using bubbling $H_2$ gas on a Pt coil electrode. Potentials were referenced to an RHE by adding 0.923 V ($0.099 + 0.059 \times pH$) in a 1 M KOH aqueous solution. For comparison, Pt/C (20 wt%, FuelCellStore; loaded on the nickel foam at $1\ mg\ cm^{-2}$) was used as an HER electrocatalyst. The impedance spectra of the electrocatalysts in a three-electrode set-up were recorded at different HER overpotentials in a 1 M KOH electrolyte. All voltages and potentials were corrected to eliminate electrolyte resistances unless noted. Electrolyte resistance: 0.94 Ω; scan rate: $1\ mV\ s^{-1}$.

**Theoretical calculations.** All computations were performed by applying the plane-wave-based DFT method with the Vienna Ab Initio Simulation Package and periodic slab models. The electron ion interaction was described with the projector augmented wave method. The electron exchange and correlation energy were treated within the generalized gradient approximation in the Perdew–Burke–Ernzerhof formalism. The cut-off energy of 400 eV and Gaussian electron smearing method with $\sigma = 0.05\ eV$ were used. The geometry optimization was performed when the convergence criterion on forces became smaller than $0.02\ eV\ Å^{-1}$ and the energy difference was $< 10^{-4}\ eV$. The adsorption energy ($E_{ads}$) of species X is calculated by $E_{ads} = E(X/slab) - E(X) - E(slab)$, and a more negative $E_{ads}$ indicates a more stable adsorption. For the DFT calculations, the reactant ($H_2O$) and intermediates (OH and H) are first adsorbed on all possible active sites of the catalyst. Afterwards, the VASP software is utilized to optimize the adsorption. For evaluating the energy barrier ($E_a = E_{TS} - E_{IS}$), the transitional state (TS) was located using the Nudged Elastic Band method. All transition states were verified by vibration analyses with only one imaginary frequency. The p($3 \times 3$)-Ni(111), p($3 \times 3$)-Mo(110), p($3 \times 3$)-$MoO_2$(110) and p($1 \times 1$)-$MoNi_4$(200) surfaces were utilized to simulate the properties of these electrocatalysts.

**Data availability.** The data that support the findings of this study are available from the corresponding author on reasonable request.

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

## Acknowledgements

This work was financially supported by the ERC Grant on 2DMATER and EC under Graphene Flagship (No. CNECT-ICT-604391). We also acknowledge the Cfaed (Center for Advancing Electronics Dresden), the Dresden Center for Nanoanalysis (DCN) at TU Dresden and Dr Horst Borrmann for the X-ray diffraction characterizations in Max Planck Institute for Chemical Physics of Solids.

## Author contributions

J.Z. and X.F. conceived and designed the experiments and wrote the paper; J.Z. carried out the synthesis and characterization of electrocatalysts; T.W. performed the DFT calculations; P.L., S.L., Z.L., X.Z., M.C. and E.Z. assisted with the HRTEM and XPS characterizations. All authors discussed the results and commented on the manuscript.

## Additional information

**Publisher's note**: 

