## [Peer Review File · Nature Communications]

Reviewers' comments:

Reviewer #1 (Remarks to the Author):

In this manuscript, MoNi₄ electrocatalyst supported by MoO₂ nanocuboids on Ni foam were prepared and their hydrogen evolution reaction performances were reported. The HER activity and durability of the derived catalyst are exceptional good, which is similar or even outperforms over Pt. The research is extensive and the methods are sound. The manuscript is well organized and well written. I recommend its publication in Nat Comm after addressing the following issues.

1. For the synthesis of the MoNi₄ electrocatalyst supported by the MoO₂ nanocuboids on nickel foam, the authors used the H₂/Ar atmosphere to calcine the as-synthesized NiMoO₄ nanocuboids to control the outward diffusion of Ni atoms. The author should state the effect of the H₂ for this synthesis process. Does it show the same result when calcined at pure Ar atmosphere?
2. For the MoNi₄/MoO₂@Ni-2h at 500 °C, the XPS spectrum confirms the molar ratio of Mo to Ni is approximately 1:1.1. What is the approximately molar ratio of MoNi₄ to MoO₂? If the molar ratio is 1:2, does it have the reduced reaction for NiMoO₄ or MoO₂ at H₂/Ar atmosphere at 500 °C? If the molar ratio of MoNi₄ to MoO₂ is not 1:2, whether have Ni or Mo existed in MoNi₄/MoO₂@Ni-2h?
3. The Characterization and investigation of electrochemical performance were carried out surpassingly. The electrochemical impedance spectroscopy of sample should be experimentally evaluated.
4. MoNi₄/MoO₂@Ni-2h shows better performance than the NiMoO₄ nanocuboids calcined for different times. Composition of those electrocatalyst is how to change.
5. Authors used the DFT calculation to study the HER mechanism of different optimized structure configuration. For the studies, how to determine the choice of Ni (111), Mo (110) and MoO₂ (110) facet?
6. Since figure 4a gives the optimized configuration of OH adsorption, I think the authors should make the corresponding explanation.

Reviewer #2 (Remarks to the Author):

The manuscript reported the synthesis and electrocatalytic performance of "MoNi₄" particles supported by MoO₂ cuboids on Ni foam (MoNi₄/MoO₂@Ni). The catalyst was prepared by the annealing NiMoO₄ cuboids grown on Ni foam at H₂/Ar atmosphere. "MoNi₄" particles were claimed to form by the outmigration of Ni in NiMoO₄ cuboid during annealing process. Although NiMoO₄ cuboids grown on Ni foam were well reported in the literatures (such as J. Mater. Chem. A, 2015, 3, 1863 etc.) and MoNi₄ was also discovered for a highly-efficient HER electrocatalysts (such as ChemElectroChem 2014, 1, 1138 etc.), there is no reports about the preparation of "MoNi₄" particles supported by MoO₂ cuboids on Ni foam through annealing NiMoO₄ cuboid precursor yet. It is interesting to note that the reported catalyst MoNi₄/MoO₂@Ni exhibited the HER activity comparable to Pt/C with a low Tafel slope of 30 mV decade⁻¹ in 1 M KOH electrolyte. If all conclusions can be well supported by the data, these results are interesting for the development of practical water-alkali electrolyzer. However, at the present status, there are quite a few unclear and critical issues which should be unambiguously resolved to support the authors' claim in this manuscript.

1. The authors claimed that the particles on MoO₂ cuboids after the annealing are MoNi₄ and attributed the high HER activity of the catalyst MoNi₄/MoO₂@Ni to "MoNi₄". The calculations also based on the assumption of these particles are "MoNi₄". However, the presented data cannot support this claims.
 - i) In Supplementary Figure 8, the authors assigned two negligible signals at 33.0° and 72.7° to the X-Ray diffraction signals of (113) and (411) planes of MoNi₄ (JCPDS, No. 03-1036), which is inconvincible. First, the two marked peaks are hardly distinguishable. Second, XRD peaks of 33.0° and 72.7° were not found in JCPDS, No. 03-1036. Third, the diffractions of (113) was not found

and the diffractions of (113) should be at 73.5° in JCPDS, No. 03-1036. Moreover, there are several more-like peaks unindexed.

ii) In Figure 2, the authors assigned the lattice distance of 0.28 nm and diffraction spot in Fig 2d (inset) to (113) planes of MoNi_4 . If the XRD pattern cannot clearly confirm that the particles are not MoNi_4 . These features could be ascribed to other crystalline structures.

iii) The authors said "the scanning TEM (STEM)-EDX characterizations indicate that the surface nanoparticles are constituted by only Mo and Ni with an atomic ratio of 1:4". Please provide the EDX spectrum of the nanoparticle. It is hard to believe that the atomic ratio is exactly "1:4".

iv) The XRD experiments with synchrotron radiation X-ray might be a better solution to collect convincing XRD data and give the crystalline structure of these nanoparticles. The TEM with higher resolution, maybe atomic resolution HRTEM, equipped with EELS would be powerful technique to reveal whether these nanoparticles are MoNi_4 .

2. If the data cannot support that the nanoparticles are MoNi_4 , the main conclusions together with the calculation based on MoNi_4 will not be correct.

3. In XRD pattern of NiMoO_4 nanocuboids on the nickel foam, several strong XRD peaks were not assigned. If these peaks were not from NiMoO_4 , the nanocuboids are not phase-pure NiMoO_4 . What are they from?

4. The authors claimed that NiMoO_4 nanocuboids decomposed into MoO_2 cuboids and MoNi_4 nanoparticles which formed by the outward diffusion of inside Ni under the condition of H_2/Ar , 500°C for 2 h. STEM-EDX mapping showed that there was no Ni in MoO_2 cuboids after annealing for 2h. However, they also claimed "In addition, with increased calcination time at 500°C , the MoNi_4 nanoparticles gradually emerged and grew into bulk particles on the MoO_2 nanocuboids (Supplementary Fig. 2)." Supplementary Fig. 2 showed much more nanoparticles formed on the outsides after 4h annealing. Where are the additional Ni from? It seems these data are contradictory.

5. The authors said "Elemental mapping, energy dispersive spectroscopy (EDS) and X-ray photoelectron spectroscopy (XPS) confirm that the NiMoO_4 nanocuboids consist of Ni, Mo and O elements and the molar ratio of Ni to Mo is approximately 1:1 (Supplementary Fig. 5-7)." However, the data in EDX spectrum (Supplementary Fig. 5e) showed that the molar ratio of Ni to Mo is NOT 1:1.

6. The authors said "The corresponding EDX analysis further confirms that the products are composed of Mo, Ni and O and the molar ratio of Mo to Ni is approximately 1:1.3 (Supplementary Fig. 9)." If the catalyst was converted from NiMoO_4 , why the molar ratio of Mo to Ni is NOT 1:1?

7. The authors provided the XPS spectra in Figure 10-13, but did not explain the XPS peaks and the chemical states in the catalyst, even in the supporting information. Besides the XPS peaks of metallic Mo^0 and Ni^0 at 229.3 eV and 852.5 eV, there are a lot other XPS peaks. What are these peaks from? How did the authors conclude "the formation of MoNi_4 " with ignoring other peaks in Mo, Ni and O XPS spectra?

8. The authors said "...the ohmic potential drop loss from the electrolyte resistance has been subtracted (Supplementary Fig. 15)." How did the authors subtract the ohmic potential drop loss from the electrolyte resistance? Supplementary Fig. 15 did not provide this information.

9. In Supplementary Fig. 15, there is a serious problem about the potential transform between standard potential (vs. RHE) and Hg/HgO reference electrode, where the calibrated data should be 99 mV, not 0.99 mV.

10. The authors said "For comparison, pure Ni nanosheets and MoO_2 nanocuboids were also prepared on the nickel foam using the same synthesis method described above for the MoNi_4 electrocatalyst (Supplementary Fig. 16-20)". The reviewer doubts that these control catalysts were prepared by using the SAME synthesis method. Supplementary Figure 16 presented SEM images of $\text{Ni}(\text{OH})_2$ nanosheets and MoO_3 nanostructures on the nickel foam. Are these the precursors for the control samples?

11. The above two control sample of Ni nanosheets and MoO_2 nanocuboids on the nickel foam are not qualified control samples, because these two control sample showed completely different morphology, adhesion, BET etc. The comparison using these two control sample will not draw the correct conclusions. If the authors could remove the " MoNi_4 " nanoparticles without damaging the MoO_2 cuboids, that would be a good control sample.

12. Furthermore, when you prepared MoO₂ control samples on Ni foam, both of hydrothermal and pyrolysis processes may act as a nickel source, causing the resulting samples are not Ni-doped MoO₂, even forming NiMo alloy.

13. In order to support authors' claim, the clear characterizations for NiMoO₄-400 °C, MoNi-600 °C etc. in Supplementary Fig. 22 should be provided.

14. In stability test (Figure 3d), the stability of the catalyst at 100, or 200 mA/cm are actually not good. The degradation happened fastly.

15. The unclear data in Supplementary Fig. 26-28 didn't support the conclusion of no structural variations after HER durability assessments.

16. Again, for the discussions about HER active center and DFT calculation, if the authors failed to prove the particles were "MoNi₄", all these data and discussion are not correct.

17. The width of "NiMoO₄" cuboids is around 1 micrometer. Why did the authors name it "nanocuboid"?

Overall, the prepared catalyst showed good HER activity, but the authors failed to clearly characterize the composition and structure of the catalyst, which could cause completely wrong conclusions. The authors did not analyze the data and design the control experiments carefully and seriously, which made that the main conclusions were not supported by the data. The reviewer do not think the manuscript is suitable for the publication in this journal, at least at its present status.

Reviewer #3 (Remarks to the Author):

The electrocatalytic hydrogen evolution reaction (HER) from water splitting process is the most economical and effective route for the future hydrogen economy. The exploration of electrocatalysts for hydrogen evolution reaction in alkaline electrolytes, however, is much more meaningful for overall water splitting. Recently, various Pt-free HER electrocatalysts with a decreased overpotential have been reported in acidic solutions. However, under alkaline conditions, the HER activities of all reported Pt-free electrocatalysts are still far lower than that for the Pt catalyst, even though Pt is not the best catalyst in alkaline solution. In this manuscript, the authors have developed a novel MoNi₄ electrocatalyst on an economical nickel foam using a controlled outward diffusion of Ni atoms. The achieved MoNi₄ electrocatalyst showed a zero onset overpotential, an extremely decreased overpotential of 15 mV at 10 mA cm⁻² and a very low Tafel slope of 30 mV decade⁻¹ in alkaline solutions. Thus, the HER activity of the MoNi₄ electrocatalyst is highly comparable with that of the Pt catalyst and outperforms those of state-of-the-art Pt-free electrocatalysts. Combined with experimental and theoretical analyses, sufficient investigations and understandings on the electrocatalyst synthesis and the fundamental HER mechanism prove that the excellent HER activity of the MoNi₄ electrocatalyst is due to the largely decreased energy barrier of the Volmer step.

In my view, the achieved performance of HER activity in the current work is outstanding and will set the new state-of-art; the results are convincing and the manuscript has been well written. Therefore, I believe that this high-level work will draw a broad attention in electrocatalysis (especially, water splitting), energy storage and materials sciences. I thus strongly recommend the publication of this work in Nature Communications after addressing the below minor issues:

1. In Figure 4c, it should be that two hydrogen atoms combine into a H₂ molecule in the Tafel step. The authors shall carefully check it.
2. I would suggest the authors to provide a detailed method for normalizing HER activity of the MoNi₄ electrocatalyst based on its surface area?
3. The Supplementary Figure 39 can be used as Figure 4c and the present Figure 4c is moved to the SI.
4. The balls with different colors should be defined in manuscript and SI.
5. The authors might refer to a closely related work about growing NiMoO₄ in Ni foam, Adv. Energy Mater. 2015, 5, 1401172.

We address the concerns of Reviewer 1# as follows:

Comments:

In this manuscript, MoNi₄ electrocatalyst supported by MoO₂ nanocuboids on Ni foam were prepared and their hydrogen evolution reaction performances were reported. The HER activity and durability of the derived catalyst are exceptional good, which is similar or even outperforms over Pt. The research is extensive and the methods are sound. The manuscript is well organized and well written. I recommend its publication in Nat. Comm. after addressing the following issues.

Questions 1:

For the synthesis of the MoNi₄ electrocatalyst supported by the MoO₂ nanocuboids on nickel foam, the authors used the H₂/Ar atmosphere to calcine the as-synthesized NiMoO₄ nanocuboids to control the outward diffusion of Ni atoms. The author should state the effect of the H₂ for this synthesis process. Does it show the same result when calcined at pure Ar atmosphere?

Response:

We thank the reviewer for the valuable comments. According to the suggestion, the NiMoO₄ cuboids were heated at 500 °C for 2 h in Ar atmosphere (NiMoO₄-Ar). The SEM, elemental mapping, EDX, XRD, and electrochemical tests were then conducted. As shown in Fig. R1, only vertically-aligned cuboids with smooth surfaces are observed. The cuboids are composed of Ni, Mo and O elements and the molar ratio of Ni to Mo is approximately 0.94:1. The XRD analysis confirms that the obtained cuboids are NiMoO₄ (Fig. R2). The NiMoO₄-Ar showed a HER overpotential of ~270 mV at a current density of 10 mA/cm², which was far higher than 15 mV for MoNi₄/MoO₂@Ni. These results manifest that H₂ gas is a key parameter for the construction of MoNi₄/MoO₂@Ni.

Accordingly, the formation mechanism of MoNi₄/MoO₂@Ni is proposed as following:

First, the reduction of MoO₄²⁻ into MoO₂ will induce the out-diffusion of Ni⁰; second, the out-diffusion of Ni⁰ and the further reduction of surface MoO₂ into Mo⁰ lead to the formation of surface MoNi₄ nanoparticles on the MoO₂ cuboids.

We have included above results in Supplementary Fig. 4-5 and 29.

Figure R1 | Morphology and chemical composition analyses of NiMoO₄-Ar. a-c) SEM images of the obtained NiMoO₄-Ar when the NiMoO₄ cuboids on the nickel foam were heated at 500 °C for 2 h in Ar atmosphere; d) Corresponding elemental mapping images of Ni, Mo and O; e) related EDX analysis.

Figure R2 | The XRD pattern of NiMoO₄-Ar when the NiMoO₄ cuboids on the nickel foam were heated at 500 °C for 2 h in Ar atmosphere.

Figure R3 | The polarization curves of NiMoO₄-Ar when the NiMoO₄ cuboids on the nickel foam were heated for 2 h in Ar and H₂/Ar atmosphere, respectively.

Questions 2:

For the MoNi₄/MoO₂@Ni-2h at 500 °C, the XPS spectrum confirms the molar ratio of Mo to Ni is approximately 1:1.1. What is the approximately molar ratio of MoNi₄ to MoO₂? If the molar ratio is 1:2, does it have the reduced reaction for NiMoO₄ or MoO₂ at H₂/Ar atmosphere at 500 °C? If the molar ratio of MoNi₄ to MoO₂ is not 1:2, whether have Ni or Mo existed in MoNi₄/MoO₂@Ni-2h?

Response:

In order to estimate the molar ratio of MoNi₄ to MoO₂, the MoNi₄/MoO₂ electrocatalysts were synthesized on carbon cloth (MoNi₄/MoO₂@C). Afterwards, the MoNi₄/MoO₂@C was immersed into a 2 M H₂SO₄ solution for 12 h at 70 °C so that the surface MoNi₄ was etched away. The morphologies before and after the etching process are shown in Fig. R4. Obviously, the surface MoNi₄ nanoparticles supported by the MoO₂ cuboids disappeared. Meanwhile, the corresponding EDX analyses reveal that the atomic ratio of Ni to Mo decreased to 0.05:1 from initial value of 1.41:1 (Fig. R5-6). These results suggested that the surface MoNi₄ nanoparticles in MoNi₄/MoO₂@C were efficiently removed. The corresponding weights of the samples are summarized in Table R1. Accordingly, the molar ratio of MoNi₄ to MoO₂ is estimated to be 0.04:1.

There is no Mo or Ni metal in the MoNi₄/MoO₂@Ni. As shown in Fig. 2c, abundant pores (mainly mesopores and micropores) are formed in supporting MoO₂ cuboids as a result of the transformation of NiMoO₄ into MoNi₄ and MoO₂. As revealed in Fig. R7, Ni also exists in the

underlying MoO_2 , which means that the MoNi_4 is also formed in the inner pores of the MoO_2 cuboids. Nevertheless, such inner MoNi_4 is difficult to be etched away. On the other hand, along with longer calcination time, the inert MoNi_4 are further moved to the surface and aggregated into bulk MoNi_4 particles (Supplementary Fig. 2).

We have added above results in Supplementary Fig. 43-46.

Figure R4 | Morphology characterization. SEM images of the $\text{MoNi}_4/\text{MoO}_2@\text{C}$: a-c) before and d-f) after etching MoNi_4 in 2 M H_2SO_4 solution.

Figure R5 | Chemical composition analyses of MoNi₄/MoO₂@C before etching MoNi₄ in acidic solution. a) SEM image; b-d) corresponding elemental mapping images of Ni, Mo and O; e) related EDX analysis.

Figure R6 | Chemical composition analyses of MoNi₄/MoO₂@C after etching MoNi₄ in acidic solution. a) SEM image; b-d) corresponding elemental mapping images of Ni, Mo and O; e) related EDX analysis.

Table R1. The content of MoNi₄ in MoNi₄/MoO₂.

		MoNi ₄ @MoO ₂	MoO ₂	MoNi ₄	MoNi ₄ :MoO ₂ (weight ratio)	MoNi ₄ :MoO ₂ (molar ratio)
Weight	Sample 1 [#]	34.52 mg	31.22 mg	3.30 mg	1.05:10	0.041:1
	Sample 2 [#]	37.22 mg	33.82 mg	3.40 mg	1.01:10	0.039:1

Figure R7 | Morphology and chemical composition analyses. TEM and corresponding elemental mapping images of MoNi₄/MoO₂@Ni.

Questions 3:

The characterization and investigation of electrochemical performance were carried out surpassingly. The electrochemical impedance spectroscopy of sample should be experimentally evaluated.

Response:

According to the reviewer's valuable suggestion, the electrochemical impedance spectroscopy (EIS) of catalysts have been measured in Ar-saturated 1 M KOH aqueous solution at -0.1 V vs. RHE with 10 mV AC potential from 10 kHz to 0.01 Hz. The recorded impedances were presented in the form of imaginary (Im) vs. real (Re) parts at various frequencies. The high frequency interception of the Re-axis represents the resistance of the electrodes. The width of the semicircle on the Re-axis corresponds to the charge-transfer resistances and indicates the overall kinetic effects. As revealed in Fig. R8, all catalysts exhibited almost the similar intrinsic resistance (~0.94 ohm), while the charge-transfer resistance of the MoNi₄/MoO₂@Ni electrocatalyst was much lower than those of the MoO₂ and Ni electrocatalysts, suggesting a faster HER kinetic process on the MoNi₄/MoO₂@Ni electrocatalyst.

We have included above EIS results in Supplementary Fig. 20.

Figure R8 | Electrochemical impedance spectroscopy (EIS) analyses of the catalysts.

Questions 4:

MoNi₄/MoO₂@Ni-2h shows better performance than the NiMoO₄ nanocuboids calcined for different times. Composition of those electrocatalyst is how to change.

Response:

As show in Fig. R9 and Supplementary Fig. 2, we utilized SEM and EDX characterizations to investigate the morphologies and chemical compositions of MoNi₄/MoO₂@Ni with different calcination times. Along with the increased calcination time from 0 to 4 h, the molar ratio of Ni to Mo is augmented from initial 0.94:1 to 3.72:1 as a result of the aggregation of MoNi₄ on the surfaces of MoO₂ cuboids.

We have included above results in Supplementary Fig. 3.

Figure R9 | Morphology and chemical composition analyses. SEM images and corresponding EDX analyses of the MoNi₄/MoO₂@Ni when NiMoO₄ nanocuboids on the nickel foam are calcined for different time at 500 °C in a H₂/Ar atmosphere: a-b) 0 h; c-d) 1 h; e-f) 2 h; g-h) 4 h.

Questions 5:

Authors used the DFT calculation to study the HER mechanism of different optimized structure configuration. For the studies, how to determine the choice of Ni (111), Mo (110) and MoO₂ (110) facet?

Response:

The HER process occurs on the exposed surfaces of electrocatalysts. As shown in Fig. 2d-f and Supplementary Fig. 11 and 25, MoO₂(110) facets show strong XRD diffraction peaks and can be observed using the HRTEM. These results indicate that the (110) facet of MoO₂ are mainly exposed. The Ni(111) and Mo(110) facets are mainly determined by the strongest XRD diffraction signals. For MoNi₄, we have also calculated energy barrier of Volmer step on the different facets including (200), (113) and (001) (Fig. R10). The energy barriers of the Volmer step on the (200) facet is about 0.39 eV, which is lower than 0.44 eV for Pt catalyst, suggesting a fast Volmer step.

Figure R10 | DFT calculations. The calculated adsorption free energy diagram for the Volmer step on different facets of MoNi₄.

Questions 6:

Since Figure 4a gives the optimized configuration of OH adsorption, I think the authors should make the corresponding explanation.

Response:

We thank the Reviewer for valuable suggestion. For the DFT calculations, the reactant (H₂O) and intermediates (OH and H) are firstly adsorbed on all possible active sites of the catalyst. Afterwards, the VASP software is utilized to optimize the adsorption. Finally, the adsorption configurations of OH with the minimum energy, (that is, the thermodynamically most stable ones) are obtained. Therefore, the structures in Figure R11 are the optimized and most stable adsorption configurations of each surface species on the catalysts.

We have included related explanation in the part of theoretical calculations in the revised manuscript (page 15).

Figure R11 | DFT calculations. The calculated free energies of OH adsorption.

We address the concerns of Reviewer 2[#] as follows:

Comments:

The manuscript reported the synthesis and electrocatalytic performance of “MoNi₄” particles supported by MoO₂ cuboids on Ni foam (MoNi₄/MoO₂@Ni). The catalyst was prepared by the annealing NiMoO₄ cuboids grown on Ni foam at H₂/Ar atmosphere. “MoNi₄” particles were claimed to form by the outmigration of Ni in NiMoO₄ cuboid during annealing process. Although NiMoO₄ cuboids grown on Ni foam were well reported in the literatures (such as *J. Mater. Chem. A*, 2015, 3, 1863 etc.) and MoNi₄ was also discovered for a highly-efficient HER electrocatalysts (such as *ChemElectroChem* 2014, 1, 1138 etc.), there is no reports about the preparation of “MoNi₄” particles supported by MoO₂ cuboids on Ni foam through annealing NiMoO₄ cuboid precursor yet. It is interesting to note that the reported catalyst MoNi₄/MoO₂@Ni exhibited the HER activity comparable to Pt/C with a low Tafel slope of 30 mV decade⁻¹ in 1 M KOH electrolyte. If all conclusions can be well supported by the data, these results are interesting for the development of practical water-alkali electrolyzer. However, at the present status, there are quite a few unclear and critical issues which should be unambiguously resolved to support the authors’ claim in this manuscript.

Response:

We greatly appreciate the reviewer for the valuable comments. In the past, the “MoNi₄” has been synthesized by electrochemical deposition or sputtering methods (*Electrochim. Acta*, 2000, **45**, 4151; *J. Appl. Electrochem.*, 1990, **20**, 32.), but the refereed “MoNi₄” normally represents a chemical composition of ~1:4 rather than a defined crystal structure. As demonstrated in the suggested literature (*ChemElectroChem* 2014, **1**, 1138), there are no crystal structure information about “MoNi₄”, such as lattice fringes in high-resolution TEM, selected-area electron diffraction (SAED) pattern or X-ray diffraction (XRD) pattern. In our work, on the basis of the Reviewer’s excellent suggestions, we have further carried out a series of structural characterizations including high-resolution TEM, selected-area electron diffraction (SAED), STEM-EDX analysis, SEM, XPS and XRD analyses. We can now confirm the structure of the MoNi₄/MoO₂@Ni. Please see below for the detail information.

Questions 1:

The authors claimed that the particles on MoO₂ cuboids after the annealing are MoNi₄ and attributed the high HER activity of the catalyst MoNi₄/MoO₂@Ni to “MoNi₄”. The calculations also

based on the assumption of these particles are “MoNi₄”. However, the presented data cannot support this claims.

i) In Supplementary Figure 8, the authors assigned two negligible signals at 31.0° and 72.7° to the X-Ray diffraction signals of (113) and (411) planes of MoNi₄ (JCPDS, No. 03-1036), which is inconvincible. First, the two marked peaks are hardly distinguishable. Second, XRD peaks of 31.0° and 72.7° were not found in JCPDS, No. 03-1036. Third, the diffractions of (113) was not found and the diffractions of (113) should be at 73.5° in JCPDS, No. 03-1036. Moreover, there are several more-like peaks unindexed.

Response:

In the previous version, referring to the literature (*Chin. Phys. B*, 2015, **24**, 108202), we ascribed the XRD diffraction peak at 31° to the (113) of MoNi₄. We are sorry for this mistake. XRD analysis has been conducted again with a more sensitive XRD instrument (SEIFERT analytical X-ray in Max Planck Institute for Chemical Physics of Solids). A stronger diffraction signal is detected at 31° (Fig. R12). According to the constructive suggestion of the Reviewer, we carefully calculated the lattice distances of (200) and (113) facets using the following equation:

$$\frac{1}{d^2} = \frac{h^2+k^2}{a^2} + \frac{l^2}{c^2}$$

(h, k, l) is the miller index of a facet and a, c and d are the side lengths in a tetragonal crystal system. The lattice distances of the (113) and (200) facets are thus calculated to be 0.11 nm and 0.28 nm, respectively. Therefore, the XRD diffraction peak at 31° should be assigned to the (200) facet rather than the (113) facet of the MoNi₄.

We have corrected the corresponding descriptions in the revised manuscript.

Figure R12 | The XRD pattern of MoNi₄/MoO₂@Ni.

ii) In Figure 2, the authors assigned the lattice distance of 0.28 nm and diffraction spot in Fig 2d (inset) to (113) planes of MoNi₄. If the XRD pattern cannot clearly confirm that the particles are not MoNi₄. These features could be ascribed to other crystalline structures.

Response:

As shown in Fig. R12, the observed XRD diffraction peak at 31° demonstrates a lattice distance of 0.28 nm, which accords well with the result from HRTEM and SEAD diffraction pattern. This facet with a lattice distance of 0.28 nm is assigned to the (200) plane of the MoNi₄.

iii) The authors said “the scanning TEM (STEM)-EDX characterizations indicate that the surface nanoparticles are constituted by only Mo and Ni with an atomic ratio of 1:4”. Please provide the EDX spectrum of the nanoparticle. It is hard to believe that the atomic ratio is exactly “1:4”.

Response:

We apologize for the round-off “1:4” in the previous version. The original TEM image and related EDX spectrum of the surface nanoparticle have been provided in Fig. R13. The atomic ratio of Mo to Ni is exactly 1:3.84.

We have corrected the related value in the revised manuscript (page 6) and Supplementary Fig. 13.

Figure R13 | The original EDX spectrum of the surface MoNi₄ nanoparticles.

iv) The XRD experiments with synchrotron radiation X-ray might be a better solution to collect convincing XRD data and give the crystalline structure of these nanoparticles. The TEM with higher resolution, maybe atomic resolution HRTEM, equipped with EELS would be powerful technique to reveal whether these nanoparticles are MoNi₄.

Response:

As shown in Fig. R12, the XRD analysis was conducted again using a more sensitive XRD instrument. The stronger diffraction peak was observed at 31°, which was attributed to the (200) facet of MoNi₄. According to the Reviewer's constructive suggestion, we carried out atomic-resolution HRTEM analysis (equipped with EELS). The corresponding EDX and EELS spectrum are provided in Fig. R14. Clearly, the surface nanoparticles are composed of Ni and Mo and the atomic ratio of Ni to Mo is about 3.89. Unfortunately, due to the extremely large thickness (up to 1 μm) and surface roughness of MoNi₄/MoO₂, atomic-resolution HRTEM image was not successfully achieved in Prof. Mingwei Chen's group (Prof. Mingwei Chen's group has the most advanced HRTEM instruments and is well known for the HRTEM characterizations). We are so sorry for the lack of atomic-resolution HRTEM image.

Figure R14 | The EDX and EELS spectrum of the surface MoNi₄ nanoparticles.

Questions 2:

If the data cannot support that the nanoparticles are MoNi₄, the main conclusions together with the calculation based on MoNi₄ will not be correct.

Response:

We fully agree with the Reviewer. At present, our available results can confirm that the surface nanoparticles are the MoNi₄. First, the TEM (STEM)-EDX spectrum confirm that the surface nanoparticles are constituted by Mo and Ni elements and the atomic ratio of Ni to Mo is 3.84:1. Second, the high-resolution TEM and SEAD pattern reveal that the lattice distance of surface nanoparticles is about 0.28 nm, which is consistent with the (200) facet of MoNi₄. Third, the XRD diffraction peak appears at 31°, which accords well with the (200) facet of MoNi₄ (JCPDS, No. 65-5480).

Questions 3:

In XRD pattern of NiMoO₄ nanocuboids on the nickel foam, several strong XRD peaks were not assigned. If these peaks were not from NiMoO₄, the nanocuboids are not phase-pure NiMoO₄. What are they from?

Response:

We are sorry that the XRD pattern of NiMoO₄ nanocuboids is not well described in the previous version. The as-obtained NiMoO₄ cuboids on the nickel foam were dried for 12 h at 80 °C in an electric oven. Thus, the crystalline hydrated H₂O molecules still exist as NiMoO₄•xH₂O. As shown in

Fig. R15, these strong peaks originate from $\text{NiMoO}_4 \cdot x\text{H}_2\text{O}$. The product consists of NiMoO_4 and $\text{NiMoO}_4 \cdot x\text{H}_2\text{O}$.

We have included the related description in Supplementary Fig. 6.

Figure R15 | The XRD pattern of the as-constructed NiMoO_4 cuboids on the nickel foam.

Questions 4:

The authors claimed that NiMoO_4 nanocuboids decomposed into MoO_2 cuboids and MoNi_4 nanoparticles which formed by the outward diffusion of inside Ni under the condition of H_2/Ar , 500 °C for 2 h. STEM-EDX mapping showed that there was no Ni in MoO_2 cuboids after annealing for 2h. However, they also claimed “In addition, with increased calcination time at 500 °C, the MoNi_4 nanoparticles gradually emerged and grew into bulk particles on the MoO_2 nanocuboids (Supplementary Fig. 2).” Supplementary Fig. 2 showed much more nanoparticles formed on the outsides after 4h annealing. Where are the additional Ni from? It seems these data are contradictory.

Response:

We are sorry for the inappropriate description in the previous manuscript. In order to well investigate the elemental distribution of $\text{MoNi}_4/\text{MoO}_2@/\text{Ni}$, STEM-EDX mapping analysis was further conducted in different areas. As we discussed in the response to Reviewer 1 (second question), a similar question has been addressed. Abundant pores (mainly mesopores and micropores) are formed in supporting MoO_2 cuboids due to the transformation of NiMoO_4 into MoNi_4 and MoO_2 . As revealed in Fig. R7, Ni also exists in underlying MoO_2 , which means that the MoNi_4 is also produced

in the inner pores of the MoO₂ besides on the surfaces. However, along with longer calcination time, the inert MoNi₄ will further moved to the surface and aggregated into bulk MoNi₄ particles (Supplementary Fig. S2-3).

We have revised the corresponding discussions in the revised manuscript (page 6).

Questions 5:

The authors said “Elemental mapping, energy dispersive spectroscopy (EDS) and X-ray photoelectron spectroscopy (XPS) confirm that the NiMoO₄ nanocuboids consist of Ni, Mo and O elements and the molar ratio of Ni to Mo is approximately 1:1 (Supplementary Fig. 5-7).” However, the data in EDX spectrum (Supplementary Fig. 5e) showed that the molar ratio of Ni to Mo is NOT 1:1.

Response:

We agree with the reviewer’s comment that the molar ratio of Ni to Mo NiMoO₄ nanocuboids is not exactly a value of 1:1. The original EDX spectrum data of the NiMoO₄ cuboids on the Ni foam is provided in Fig. R16. The accurate atomic ratio of Ni to Mo is 1.06:1, which is very close to 1:1. The negligible excess of Ni is from the nickel foam.

Map Scan Spectrum	Line Type	Apparent Concentration	k Ratio	Wt%	Wt% Sigma	Atomic %	Standard Label	Factory Standard	Standard Calibration Date
C	K series	1.17	0.01168	3.79	0.44	10.51	Pure Element	Yes	
O	K series	14.93	0.11080	28.98	0.33	60.14	SiO2	Yes	
Ni	K series	49.90	0.49698	26.40	0.42	14.98	Ni	Yes	
Mo	L series	54.79	0.54795	40.85	0.42	14.18	Pure Element	Yes	
Total				100.00		100.00			

Figure R16 | The original EDX spectrum data of the as-constructed NiMoO₄ cuboids on the nickel foam.

Questions 6:

The authors said “The corresponding EDX analysis further confirms that the products are composed of Mo, Ni and O and the molar ratio of Mo to Ni is approximately 1:1.3 (Supplementary Fig. 9).” If the catalyst was converted from NiMoO₄, why the molar ratio of Mo to Ni is NOT 1:1?

Response:

We are sorry for the unclear description in the previous version. Although the MoNi₄/MoO₂@Ni catalyst was converted from NiMoO₄ cuboids on the nickel foam, the molar ratio of surface Mo to Ni in the MoNi₄/MoO₂@Ni catalyst should be not 1:1 as a result of the existence of surface MoNi₄ nanoparticles. The corresponding SEM-EDX analysis can reflect the elemental composition of

catalyst surfaces with a maximum depth of several hundreds of nanometers, which depends on the applied voltage and the intrinsic property of samples. Here, the applied voltage was as low as 3 kV. Therefore, the molar ratio of surface Mo to Ni is dependent on the content of surface MoNi_4 and MoO_2 . As shown in Fig. R 17 and Fig. 2c, the surface of $\text{MoNi}_4/\text{MoO}_2@Ni$ is constituted by plentiful MoNi_4 nanoparticles and underlying MoO_2 so that the detected molar ratio of Mo to Ni is 1:1.3 rather than 1:1.

Figure R17 | The structural scheme of the $\text{MoNi}_4/\text{MoO}_2$ cuboids.

Questions 7:

The authors provided the XPS spectra in Figure 10-13, but did not explain the XPS peaks and the chemical states in the catalyst, even in the supporting information. Besides the XPS peaks of metallic Mo^0 and Ni^0 at 229.3 eV and 852.5 eV, there are a lot other XPS peaks. What are these peaks from? How did the authors conclude “the formation of MoNi_4 ” with ignoring other peaks in Mo, Ni and O XPS spectra?

Response:

We are sorry for the reckless description of XPS results in the previous version. Based on the XPS results, we only confirm the existence of Mo^0 and Ni^0 rather than MoNi_4 on the surfaces of the $\text{MoNi}_4/\text{MoO}_2@Ni$ catalyst. The high-resolution XPS spectrum of O 1s, Ni 2p and Mo 3d are shown in Fig. R18. The O 1s peaks at 528.5 eV, 530.3 eV and 537.2 eV are assigned to Ni-O, Mo-O and H-O-H in NiMoO_4 cuboids, respectively (Fig. R18a). The O 1s signal of Mo-O in $\text{MoNi}_4/\text{MoO}_2$ appears at 531.1 eV. As revealed in Fig. R18b, the peaks at 855.8 eV, 861.7 eV, 873.7 eV and 881.4 eV are ascribed to Ni^{2+} 2p_{3/2}, Ni^{2+} 2p_{3/2} satellite, Ni^{2+} 2p_{1/2} and Ni^{2+} 2p_{1/2} satellite, respectively. Ni^0 2p_{3/2}, Ni^0 2p_{3/2} satellite, Ni^0 2p_{1/2} and Ni^0 2p_{1/2} satellite are observed at 852.6 eV, 858.5 eV, 869.5 eV and 876.0 eV, respectively. The peaks located at 230.2 eV, 232.5 eV, and 235.4 eV in Fig. R18c belong to Mo^{4+} 3d_{5/2}, Mo^{4+} 3d_{3/2} and Mo^{6+} 3d_{3/2}, respectively. The Mo^0 3d_{5/2} and Mo^0 3d_{3/2} are detected at 229.3 eV and 231.5 eV, respectively. These XPS results thus confirm the existence of Mo^0 and Ni^0 in the surfaces of the $\text{MoNi}_4/\text{MoO}_2@Ni$.

We have included the above discussions in the Supplementary Fig. 15-17 and revised the corresponding description in the revised manuscript (page 6).

Figure R18 | The high-resolution XPS pattern of (a) O 1s, (b) Ni 2p and (c) Mo 3d for the NiMoO₄ cuboids, MoO₂ cuboids, Ni nanosheets and MoNi₄/MoO₂@Ni on the nickel foam.

Questions 8:

The authors said “...the ohmic potential drop loss from the electrolyte resistance has been subtracted (Supplementary Fig. 15).” How did the authors subtract the ohmic potential drop loss from the electrolyte resistance? Supplementary Fig. 15 did not provide this information.

Response:

The electrochemical impedance spectroscopy (EIS) of catalysts have been measured in Ar-saturated 1 M KOH aqueous solution at -0.1 V vs. RHE with 10 mV AC potential from 10 kHz to 0.01 Hz. The recorded impedances were presented in the form of imaginary (Im) vs. real (Re) parts at various frequencies. The high frequency interception of the Re-axis represents the resistance of the electrodes (R_s). The width of the semicircle on the Re-axis corresponds to the charge-transfer resistances and indicates the overall kinetic effects. As indicated in Fig. R8, the MoNi₄/MoO₂@Ni has an electrolyte resistance of ~0.94 ohm. The ohmic potential drop (iR_s) loss from the electrolyte resistance was then subtracted according to the following equation:

$$P_{\text{vs. RHE}} = P_{\text{vs. Hg/HgO}} + 0.059 \times \text{pH} - i \times R_s$$

$P_{vs. RHE}$ is the potential versus standard hydrogen electrode. $P_{vs. Hg/HgO}$ is the measured potential with Hg/HgO electrode as reference electrode. R_s is the electrolyte resistance. The pH value of 1 M KOH aqueous solution is about 13.97. i is the recorded current density. The polarization curves of the $MoNi_4/MoO_2@Ni$ before and after the iR_s correction are shown in Fig. R 19.

We have included the above discussion in Supplementary Fig. 20.

Figure R19 | The polarization curves of the $MoNi_4/MoO_2@Ni$ before and after the iR_s correction.

Questions 9:

In Supplementary Fig. 15, there is a serious problem about the potential transform between standard potential (vs. RHE) and Hg/HgO reference electrode, where the calibrated data should be 99 mV, not 0.99 mV.

Response:

We apologize for this writing mistake. The potential (vs. RHE) of Hg/HgO reference electrode is determined to be 0.099 V. This mistake has been corrected in Supplementary Fig. 19.

Questions 10:

The authors said “For comparison, pure Ni nanosheets and MoO_2 nanocuboids were also prepared on the nickel foam using the same synthesis method described above for the MoN_4 electrocatalyst (Supplementary Fig. 16-20).”. The reviewer doubts that these control catalysts were prepared by using the SAME synthesis method. Supplementary Figure 16 presented SEM images of $Ni(OH)_2$

nanosheets and MoO₃ nanostructures on the nickel foam. Are these the precursors for the control samples?

Response:

These Ni(OH)₂ nanosheets and MoO₃ nanostructures serve as precursors for the synthesis of pure Ni nanosheets and MoO₂ nanocuboids. We are sorry for the misleading word. The “same” just means the hydrothermal reaction conditions including reaction temperature and time. The “same synthesis method described above for the MoNi₄ electrocatalyst” has been revised to “hydrothermal reactions” in the revised manuscript (page 7). The detail information of the hydrothermal reactions is summarized in Table R2.

Table R2 | The hydrothermal reaction conditions.

	Ni(NO ₃) ₂ •6H ₂ O	(NH ₄) ₆ Mo ₇ O ₂₄ •4H ₂ O	H ₂ O	Reaction temperature	Reaction Time
NiMoO ₄ cuboids	174.4 mg	208 mg	15 mL	150 °C	6 h
Ni(OH) ₂ nanosheets	174.4 mg	—	15 mL	150 °C	6 h
MoO ₃ nanostructures	—	208 mg	15 mL	150 °C	6 h

Questions 11:

The above two control sample of Ni nanosheets and MoO₂ nanocuboids on the nickel foam are not qualified control samples, because these two control sample showed completely different morphology, adhesion, BET etc. The comparison using these two control sample will not draw the correct conclusions. If the authors could remove the “MoNi₄” nanoparticles without damaging the MoO₂ cuboids, that would be a good control sample.

Response:

We thank the Reviewer for the valuable suggestion. We have tried to synthesize the Ni cuboids and MoO₂ cuboids on the nickel foam by tuning the hydrothermal reaction conditions. Unfortunately, the cuboid-like morphology could be not realized for the Ni and MoO₂. According to the suggestion, we synthesized the MoNi₄/MoO₂ cuboids on carbon cloth (MoNi₄/MoO₂@C) (Fig. R4a-c). Then, the surface MoNi₄ nanoparticles supported by MoO₂ cuboids were etched away by immersing the MoNi₄/MoO₂@C in 2 M H₂SO₄ for 12 h at 70 °C (Fig. R4d-f). As indicated in Fig. R20, the overpotential of MoNi₄/MoO₂@C at 10 mA/cm² is largely increased to 194 mV from 53 mV after etching the

surface MoNi₄ nanoparticles away. This result prove that the surface MoNi₄ nanoparticles are responsible for the excellent HER activity of the MoNi₄/MoO₂@C.

We have included above results in the revised manuscript and Supplementary part (page 11 and Supplementary Fig. 43-46).

Figure R20 | The polarization curves of the MoNi₄/MoO₂@C before and after etching the MoNi₄ in 2 M H₂SO₄ solution.

Questions 12:

Furthermore, when you prepared MoO₂ control samples on Ni foam, both of hydrothermal and pyrolysis processes may act as a nickel source, causing the resulting samples are not Ni-doped MoO₂, even forming NiMo alloy.

Response:

We agree with the Reviewer. In order to avoid the influence of Ni foam and probe the intrinsic HER activity, we thus prepared MoO₂ nanosheets and MoNi₄/MoO₂ cuboids on the carbon cloth (MoNi₄/MoO₂@C). For comparison, the surface MoNi₄ nanoparticles of MoNi₄/MoO₂@C were further etched away in 2 M H₂SO₄ at 70 °C.

Questions 13:

In order to support authors' claim, the clear characterizations for NiMoO₄-400 °C, MoNi-600 °C etc. in Supplementary Fig. 22 should be provided.

Response:

We are sorry for the lack of XRD characterizations for NiMoO₄-400 °C and MoNi-600 °C. The XRD patterns of the NiMoO₄-400 °C and MoNi-600 °C are provided in Fig. R21. The product at 400 °C is NiMoO₄ cuboids while the product at 600 °C is MoNi₃ nanoparticles supported by MoO₂ cuboids.

We have included these results and revised the corresponding description in the revised manuscript and Supplementary part (Supplementary Fig. 27).

Figure R21 | The XRD patterns of the products after the calcination of NiMoO₄ cuboids at different temperatures in H₂/Ar atmosphere: a) 400 °C and b) 600 °C.

Questions 14:

In stability test (Figure 3d), the stability of the catalyst at 100, or 200 mA cm⁻² are actually not good. The degradation happened fastly.

Response:

As disclosed in Fig. R22, for the previous stability tests, the immersed depth of MoNi₄/MoO₂@Ni electrode was 1 cm. When the HER was carried out at a high current density (>100 mA/cm²), the excessive consumption of H₂O led to the reduction of immersed electrode area. Meanwhile, the electrolyte resistance was largely increased due to the generation of plentiful bubbles in the glass tube. These two parameters result in the degradation in the previous

stability tests. Based on the value comment, we conducted the stability tests in an upgraded system again. As show in Fig. R23, the overpotential augment of the MoNi₄/MoO₂@Ni at 200 mA/cm² is only 5 mV after a period of 10 h of hydrogen production. Therefore, the MoNi₄/MoO₂@Ni manifests an excellent stability.

We have revised the Fig. 3d in the revised manuscript (page 7).

Figure R22 | The three-electrode system for electrocatalytic HER stability tests.

Figure R23 | Long-term stability tests of the MoNi₄ electrocatalyst at different current densities: 10, 100 and 200 mA cm⁻².

Questions 15:

The unclear data in Supplementary Fig. 26-28 didn't support the conclusion of no structural variations after HER durability assessments.

Response:

After a long-term stability test, the MoNi₄/MoO₂@Ni was quickly washed using 0.1 M HCl aqueous solution and distilled water to remove the adsorbed KOH electrolyte. Then, SEM, HR-TEM, STEM-EDX analyses were conducted again. Clearly, MoNi₄ nanoparticles remained on the MoO₂ cuboids as shown in Fig. R24b. The lattice distance of ~0.28 nm belongs to the (200) facet of MoNi₄ (Fig. R24d). Fig. R25 reveals that the remaining nanoparticles are constituted by Mo and Ni element and the atomic ratio of Ni to Mo is about 3.21:1 due to the existence of supporting MoO₂ in the detected

area. These results demonstrate the structural stability of the MoNi₄/MoO₂@Ni during the HER process.

We have included above results in Supplementary Fig. 33-36.

Figure R24 | (a and b) SEM and (c and d) TEM images of MoNi₄/MoO₂@Ni after a long-term HER stability test in a 1 M KOH electrolyte.

Figure R25 | TEM and corresponding elemental mapping images of MoNi₄/MoO₂@Ni after a long-term stability test.

Questions 16:

Again, for the discussions about HER active center and DFT calculation, if the authors failed to prove the particles were “MoNi₄”, all these data and discussion are not correct.

Response:

As we discussed above, a series of structural characterizations have been further conducted. First, the TEM (STEM)-EDX spectrum confirm the surface nanoparticles are constituted by Mo and Ni elements and the atomic ratio of Ni to Mo is 3.84:1, which well approaches to 4:1. Second, the high-resolution TEM and SEAD pattern reveal that the lattice distance of surface nanoparticles is about 0.28 nm, which is consistent with the (200) facet of MoNi₄. Third, the XRD diffraction peak appears at 31°, which accords well with the (200) facet of MoNi₄ (JCPDS, No. 65-5480). Therefore, the surface nanoparticles have been proved to be the MoNi₄.

Accordingly, the HER process on the (200) facet of MoNi₄ has been further calculated by the DFT calculations. As revealed in Fig. R8, the energy barrier of Volmer step is only 0.39 eV, which is lower than 0.44 eV for the Pt.

Questions 17:

The width of “NiMoO₄” cuboids is around 1 micrometer. Why did the authors name it “nanocuboid”?

Response:

According to the suggestion, “nanocuboids” in the manuscript and Supplementary part has been revised to “cuboids”.

Comments:

Overall, the prepared catalyst showed good HER activity, but the authors failed to clearly characterize the composition and structure of the catalyst, which could cause completely wrong conclusions. The authors did not analyze the data and design the control experiments carefully and seriously, which made that the main conclusions were not supported by the data. The reviewer do not think the manuscript is suitable for the publication in this journal, at least at its present status.

Response:

On basis of the Reviewer's constructive comments and suggestions, we have carefully and seriously analyzed the data and designed the experiments again. Then, we further carried out a series of structural characterizations including HRTEM, STEM-EDX, STEM-EELS, SEAD, FESEM, elemental mapping, XRD and XPS. The surface nanoparticles and underlying cuboids have been confirmed to be MoNi₄ and MoO₂, respectively. Combined by electrochemical analyses and DFT calculations, the energy barrier of Volmer step was largely decreased to 0.39 eV on the (200) facets of MoNi₄, which is responsible for the excellent HER activity of MoNi₄/MoO₂@Ni. The updated data can fully support the conclusion.

We address the concerns of Reviewer 3[#] as follows:

Comments:

The electrocatalytic hydrogen evolution reaction (HER) from water splitting process is the most economical and effective route for the future hydrogen economy. The exploration of electrocatalysts for hydrogen evolution reaction in alkaline electrolytes, however, is much more meaningful for overall water splitting. Recently, various Pt-free HER electrocatalysts with a decreased overpotential have been reported in acidic solutions. However, under alkaline conditions, the HER activities of all reported Pt-free electrocatalysts are still far lower than that for the Pt catalyst, even though Pt is not the best catalyst in alkaline solution. In this manuscript, the authors have developed a novel MoNi₄ electrocatalyst on an economical nickel foam using a controlled outward diffusion of Ni atoms. The achieved MoNi₄ electrocatalyst showed a zero onset overpotential, an extremely decreased overpotential of 15 mV at 10 mA cm⁻² and a very low Tafel slope of 30 mV decade⁻¹ in alkaline solutions. Thus, the HER activity of the MoNi₄ electrocatalyst is highly comparable with that of the Pt catalyst and outperforms those of state-of-the-art Pt-free electrocatalysts. Combined with experimental and theoretical analyses, sufficient investigations and understandings on the electrocatalyst synthesis and the fundamental HER mechanism prove that the excellent HER activity of the MoNi₄ electrocatalyst is due to the largely decreased energy barrier of the Volmer step. In my view, the achieved performance of HER activity in the current work is outstanding and will set the new state-of-art; the results are convincing and the manuscript has been well written. Therefore, I believe that this high-level work will draw a broad attention in electrocatalysis (especially, water splitting), energy storage and materials sciences. I thus strongly recommend the publication of this work in Nature Communications after addressing the below minor issues:

Questions 1:

In Figure 4c, it should be that two hydrogen atoms combine into a H₂ molecule in the Tafel step. The authors shall carefully check it.

Response:

We appreciate the valuable comments from the reviewer. We are sorry for this mistake. Here one hydrogen atom has been added in Supplementary Fig. 52.

Questions 2:

I would suggest the authors to provide a detailed method for normalizing HER activity of the MoNi₄ electrocatalyst based on its surface area?

Response:

First, the specific surface area of MoNi₄/MoO₂ cuboids was determined to be 32 m²/g by the Brunauer-Emmett-Teller (BET) method. Second, the weight content of MoNi₄ in MoNi₄/MoO₂ was estimated to be 9.5 wt% and the loading weight of MoNi₄/MoO₂ on nickel foam was approximately 43.4 mg. Thus, the surface area of MoNi₄ supported by the MoO₂ cuboids was determined to be 0.1319 m². Finally, the HER current density of MoNi₄ is normalized with the surface area of 0.1319 m² and the obtained polarization curves are shown in Fig. R26.

Figure R26 | The normalized polarization curves of MoNi₄/MoO₂@Ni and the Pt catalyst on the nickel foam.

Questions 3:

The Supplementary Figure 39 can be used as Figure 4c and the present Figure 4c is moved to the SI.

Response:

The calculated adsorption free energy diagram for the Tafel step has been revised as Fig. 4c The proposed HER mechanism on the MoNi₄ electrocatalyst has been moved to Supplementary part (Supplementary Fig. 52).

Questions 4:

The balls with different colors should be defined in manuscript and SI.

Response:

The balls in the manuscript and Supplementary part have been defined (Fig. 4a and Supplementary Fig. 52).

Questions 5:

The authors might refer to a closely related work about growing NiMoO₄ in Ni foam, Adv. Energy Mater. 2015, 5, 1401172.

Response:

This literature has been cited as reference 46 in the manuscript (page 13).

Reviewers' comments:

Reviewer #1 (Remarks to the Author):

The authors addressed all my comments properly and made necessary revision. I would like to recommend the publication on Nat Comm as its current version.

Reviewer #2 (Remarks to the Author):

I am glad to see the authors have supplemented the necessary data to support their conclusions and correct some of the related description. Most of my concerns have been responded except for the following issues.

1. In the original manuscript, the author used the XRD peak at 33° to assign the existence of MoNi₄, however, they reproduced my comments on this peak by SILENTLY changing 33 to 31° without any note/explanation and answer the question with the newly measured "peak" at 31°. This is not right.
2. The newly recorded XRD "peak" at 31° in Fig. R12 looks like a noise spike or something artificial rather than a XRD peak, which is not reasonable. The authors used this XRD "peak" to assign the MoNi₄ as well as the lattice fringes in Fig. 3d, which is inconvincible. If XRD technique can detect the crystalline structure of the prepared MoNi₄, XRD pattern should present a set of diffraction peaks instead of one spike although they might be weak.
3. Again, if authors want to use XRD technique to support their claims about MoNi₄, the synchrotron radiation X-ray diffraction might be a better solution to collect convincible XRD data and give the crystalline structure of these nanoparticles.
4. In the response 3, the authors tried to use NiMoO₄·xH₂O to attribute the XRD peaks of the "as-constructed NiMoO₄". As clearly seen in the Figure R15, there are still quite a few strong XRD peaks cannot be indexed to either NiMoO₄ or NiMoO₄·xH₂O. Actually, most of the main XRD peaks cannot be well indexed to the referred pattern.
5. In the response 5, the authors provided the original EDX data with the atomic ratio of Ni to Mo of 1.06:1. Actually, this should not be the original data for the supplementary figure 5 in the previous version. You can clearly read that the ratio of Ni to Mo in the previous version was around 1.22 by transforming the wt% to at%. Although the authors deleted the original data set in this figure (supplementary figure 9 in the revision) and use the number of 1.06:1 instead, they should NOT just pick up the data set that did not belong to the presented EDX spectrum.
6. the reference JCPDS card should also be included in the XRD pattern for the sample NiMoO₄-400° and NiMoO₄-600° to support the conclusions

Therefore, the review thinks that authors made the significant modifications in the revision but should more seriously deal with the data and address the above-mentioned issues.

Reviewer #3 (Remarks to the Author):

In this revised version, the authors have adequately addressed my concerns. The manuscript can be accepted.

We address the concerns of Reviewer 1[#] as follows:

Comments:

The authors addressed all my comments properly and made necessary revision. I would like to recommend the publication on Nat. Comm. as its current version.

Response:

We greatly appreciate the reviewer for the positive and valuable comments.

We address the concerns of Reviewer 2[#] as follows:

Comments:

I am glad to see the authors have supplemented the necessary data to support their conclusions and correct some of the related description. Most of my concerns have been responded except for the following issues.

Response:

We greatly appreciate the reviewer's positive comments and valuable suggestions which help us to further improve the manuscript.

Question 1:

In the original manuscript, the author used the XRD peak at 33° to assign the existence of MoNi_4 , however, they reproduced my comments on this peak by SILENTLY changing 33 to 31° without any note/explanation and answer the question with the newly measured "peak" at 31° . This is not right.

Response:

We are sorry for this misunderstanding. As demonstrated by the reviewer, the observed signal peak was weak and broad (from ~ 29.9 to 33.2°) in the first draft version so that the diffraction angle was estimated to be 33° by referring to the reported value in the literature (*Chin. Phys. B*, 2015, 24, 108202) (Figure R1a). Later, according to reviewer's valuable suggestion and the analysis of re-measured XRD (using a more sensitive XRD instrument at the Max Planck Institute for Chemical Physics of Solids) in the second draft version, we corrected the initial " 33° " to the accurate " 31° " on the basis of the observed diffraction peak (Figure R1b).

Figure R1. The measured XRD patterns of the MoNi₄/MoO₂@Ni in (a) the first draft version and (b) second draft version.

Question 2:

The newly recorded XRD “peak” at 31° in Fig. R12 looks like a noise spike or something artificial rather than a XRD peak, which is not reasonable. The authors used this XRD “peak” to assign the MoNi₄ as well as the lattice fringes in Fig. 3d, which is inconvincible. If XRD technique can detect the crystalline structure of the prepared MoNi₄, XRD pattern should present a set of diffraction peaks instead of one spike although they might be weak.

Response:

Although we have conducted the XRD measurements using different instruments from different institutions (TU Dresden, Max Planck Institute for Chemical Physics of Solids, Shanghai Jiao Tong University), a set of perfect diffraction peaks of surface MoNi₄ nanoparticles was not achieved in the previous versions. Nevertheless, all XRD results in our work are original data rather than artificial data.

Actually, the disturbance of strong diffraction intensity of nickel foam and MoO₂ cuboids (especially nickel foam) resulted in the difficulty in distinguishing a set of diffraction peaks of surface MoNi₄ nanoparticles. In addition, compared to the diffraction intensity of the MoNi₄ (121) facet in the reference JCPDS card, the diffraction intensity of the (200) and (312) facets are also very weak. As a result, only a weak and broad diffraction peak of the MoNi₄ (200) facet was observed in the previous XRD measurements. In order to eliminate the disturbance of the Ni foam and to obtain enhanced diffraction peaks of surface MoNi₄ nanoparticles, the MoNi₄/MoO₂ cuboids were synthesized on carbon cloth (MoNi₄/MoO₂@C) for the XRD study. As shown in Figure R2, the weak and broad (200) and (312) facets of the MoNi₄ were identified at 31.0° and 74.7°, which are similar

to our previous XRD results. The weak signals of the MoNi₄ (200) and (312) facets are attributed to the low content of MoNi₄ in the MoNi₄/MoO₂ cuboids. Importantly, a new and strong peak at 43.5° was detected and assigned to the characteristic (121) facet of the MoNi₄ (PDF- 65-5480), which was not distinguished in the previous XRD measurements due to the disturbance of the Ni (111) facet. The photograph of the original XRD spectrum is shown in Figure R3.

We have revised the manuscript (page 6) and included this new result in Supplementary Figure 11 in the revised version.

Figure R 2. The XRD pattern of the as-synthesized MoNi₄/MoO₂ cuboids on the carbon cloth.

Figure R3. The photograph of the original XRD spectrum.

Question 3:

Again, if authors want to use XRD technique to support their claims about MoNi₄, the synchrotron radiation X-ray diffraction might be a better solution to collect convincing XRD data and give the crystalline structure of these nanoparticles.

Response:

We thank the reviewer for the valuable suggestion. Indeed, we had tried to analyze the structure of the MoNi₄/MoO₂@Ni using the synchrotron radiation X-ray diffraction in the Helmholtz Center Berlin. Unfortunately, we still could not achieve the distinguished signals of the MoNi₄, which is similar to the case of XRD analysis. Based on our present results and the literature reports (K. Codling, *et al.*, Synchrotron radiation: techniques and applications, *Springer Science & Business Media*, **2013**, Vol. 10; C. Garino, *et al.*, *Coord. Chem. Rev.*, **2014**, 277, 130.), the reason for this can be due to that the diffraction intensity of the nickel foam and the MoO₂ cuboids are also strengthened when the diffraction signals of surface MoNi₄ nanoparticles are increased.

As we discussed above, the difficulty in distinguishing a set of diffraction peaks of surface MoNi₄ nanoparticles is mainly caused by the disturbance of strong diffraction intensity of nickel foam and MoO₂ cuboids (especially nickel foam). Therefore, we synthesized the MoNi₄/MoO₂ cuboids on carbon cloth (MoNi₄/MoO₂@C), which allowed us to identify the strong characteristic peak of the MoNi₄ (121) facet and weak diffraction peaks of MoNi₄ (200) and (321) facets.

Question 4:

In the response 3, the authors tried to use NiMoO₄·xH₂O to attribute the XRD peaks of the “as-constructed NiMoO₄. As clearly seen in the Figure R15, there are still quite a few strong XRD peaks cannot be indexed to either NiMoO₄ or NiMoO₄·xH₂O. Actually, most of the main XRD peaks cannot be well indexed to the referred pattern.

Response:

As manifested in the previous draft version, most of the XRD peaks of freshly-prepared NiMoO₄ cuboids on the nickel foam (NiMoO₄@Ni) are attributed to the NiMoO₄·xH₂O (PDF-13-0128). However, as a result of uncertain number of crystalline water molecules in the NiMoO₄·xH₂O, some as-detected XRD peaks deviate from those standard peaks of the NiMoO₄·xH₂O (PDF-13-0128). In order to remove the crystalline water from NiMoO₄·xH₂O, the initial NiMoO₄@Ni was heated at

increased temperature from 200 to 400 °C in Ar atmosphere. Clearly, the XRD diffraction peaks of the obtained sample at 400 °C accord perfectly with the standard peaks of the NiMoO₄ (PDF-33-0948). This result suggests that the crystalline water molecules have been removed (Figure R4). Therefore, the XRD peaks of the initial NiMoO₄@Ni are attributed to the NiMoO₄·xH₂O and the as-constructed sample on the nickel foam is NiMoO₄ cuboids.

We have included this result in Supplementary Figure 6 in the revised version.

Figure R4. The XRD patterns of the NiMoO₄ on the nickel foam after treatment at different temperatures in Ar atmosphere.

Question 5:

In the response 5, the authors provided the original EDX data with the atomic ratio of Ni to Mo of 1.06:1. Actually, this should not be the original data for the supplementary figure 5 in the previous version. You can clearly read that the ratio of Ni to Mo in the previous version was around 1.22 by transforming the wt% to at%. Although the authors deleted the original data set in this figure (supplementary figure 9 in the revision) and use the number of 1.06:1 instead, they should NOT just pick up the data set that did not belong to the presented EDX spectrum.

Response:

We are sorry for the misunderstanding of SEM-EDX information in the previous draft version. The derived ratio of Ni to Mo of 1.22 is due to the bottom nickel foam that was included in the scanning spot. In order to get more accurate SEM-EDX information, we have conducted the SEM-EDX analysis on five different spots of NiMoO₄ cuboids on nickel foam (NiMoO₄@Ni). As revealed in Figure R5 and Table R1, the average atomic ratio of Ni to Mo is about 1: 1.01.

We have revised the manuscript (page 6) and included this result in Supplementary Figure. 9 in the revised version.

Figure R5. SEM images and corresponding EDX spectra at different spots of the NiMoO₄@Ni.

Table R1. The atomic content of Ni and Mo elements in the NiMoO₄@Ni.

	Ni (atomic content)	Mo (atomic content)	Ni:Mo (atomic ratio)	Average (atomic ratio)
b	17.6	18.9	1:1.07	
d	17.8	18.8	1:1.06	
f	16.9	16.8	1:0.99	1:1.01
g	16.0	15.9	1:0.99	
j	15.5	14.9	1:0.96	

Question 6:

the reference JCPDS card should also be included in the XRD pattern for the sample NiMoO₄-400° and NiMoO₄-600° to support the conclusions.

Response:

According to the reviewer's suggestion, the standard JCPDS cards have been included in the XRD patterns in Figure R6.

We have revised the Supplementary Figure S27 in the revised version.

Figure R6. The XRD patterns of the samples after the calcination of NiMoO₄ cuboids at different temperatures in H₂/Ar atmosphere: a) 400 °C and b) 600 °C.

Comments:

Therefore, the review thinks that authors made the significant modifications in the revision but should more seriously deal with the data and address the above-mentioned issues.

Response:

We appreciate the reviewer's positive comments on our manuscript. According to his/her valuable suggestions, we have seriously address the critical issues.

We address the concerns of Reviewer 3[#] as follows:

Comments:

In this revised version, the authors have adequately addressed my concerns. The manuscript can be accepted.

Response:

We greatly appreciate the reviewer for the positive and valuable comments.

REVIEWERS' COMMENTS:

Reviewer #2 (Remarks to the Author):

The authors basically responded my concerns. The reviewer understands the difficulty in 100% unambiguously determining the crystalline structure of MoNi₄ nanoparticles in the catalysts. The manuscript is now publishable.